# Soothing the Itch: The Role of Medicinal Plants in Alleviating Pruritus in Palliative Care

**DOI:** 10.3390/plants13243515

**Published:** 2024-12-16

**Authors:** Sara Gonçalves, Lisete Fernandes, Ana Caramelo, Maria Martins, Tânia Rodrigues, Rita S. Matos

**Affiliations:** 1Academic Clinical Center of Trás-os-Montes and Alto Douro (CACTMAD), University of Trás-os-Montes and Alto Douro, 5000-801 Vila Real, Portugal; 2School of Health, University of Trás-os-Montes and Alto Douro, 5000-801 Vila Real, Portugal; 3Associação Internacional de Aromaterapeutas Profissionais (IAAP-Portugal), 4445-088 Alfena, Portugal; taniarodrigues.soul.t@gmail.com; 4Centro de Química-Vila Real (CQ-VR), UME-CIDE Unidade de Microscopia Eletrónica-Centro de Investigação e Desenvolvimento, University of Trás-os-Montes and Alto Douro, 5000-801 Vila Real, Portugal; 5Centre for Research and Technology of Agro-Environmental and Biological Sciences (CITAB), Institute for Innovation, Capacity Building and Sustainability of Agri-Food Production (Inov4Agro), University of Trás-os-Montes and Alto Douro, 5000-801 Vila Real, Portugal; 6RISE-Health Research Network, Faculty of Medicine, University of Porto, 4200-319 Porto, Portugal; 7Palliative Medicine, Local Health Unit of Trás-os-Montes and Alto Douro EPE, 5400-261 Chaves, Portugal; 8Local Health Unit of Trás-os-Montes and Alto Douro (ULSTMAD), 5050-275 Peso da Régua, Portugal; 9Palliative Medicine, Local Health Unit of Nordeste, 5370-210 Mirandela, Portugal

**Keywords:** medicinal plants, chamomile, licorice, lavender, peppermint, evening primrose, curcuma, aloe vera, calendula, pruritus

## Abstract

Chronic pruritus, or persistent itching, is a debilitating condition that severely impacts quality of life, especially in palliative care settings. Traditional treatments often fail to provide adequate relief or are associated with significant side effects, prompting interest in alternative therapies. This review investigates the antipruritic potential of eight medicinal plants: chamomile (*Matricaria chamomilla*), aloe vera (*Aloe barbadensis*), calendula (*Calendula officinalis*), curcumin (*Curcuma longa*), lavender (*Lavandula angustifolia*), licorice (*Glycyrrhiza glabra*), peppermint (*Mentha piperita*), and evening primrose (*Oenothera biennis*). These plants are analyzed for their traditional applications, active bioactive compounds, mechanisms of action, clinical evidence, usage, dosage, and safety profiles. Comprehensive searches were conducted in databases including PubMed, Web of Science, Scopus, and b-on, focusing on in vitro, animal, and clinical studies using keywords like “plant”, “extract”, and “pruritus”. Studies were included regardless of publication date and limited to English-language articles. Findings indicate that active compounds such as polysaccharides in aloe vera, curcuminoids in turmeric, and menthol in peppermint exhibit significant anti-inflammatory, antioxidant, and immune-modulating properties. Chamomile and calendula alleviate itching through anti-inflammatory and skin-soothing effects, while lavender and licorice offer antimicrobial benefits alongside antipruritic relief. Evening primrose, rich in gamma-linolenic acid, is effective in atopic dermatitis-related itching. Despite promising preclinical and clinical results, challenges remain in standardizing dosages and formulations. The review highlights the necessity of further clinical trials to ensure efficacy and safety, advocating for integrating these botanical therapies into complementary palliative care practices. Such approaches emphasize holistic treatment, addressing chronic pruritus’s physical and emotional burden, thereby enhancing patient well-being.

## 1. Introduction

Medicinal plants have been essential to healthcare systems across cultures and time, forming the basis of numerous traditional and modern therapeutic practices [1]. These plants, revered for their diverse pharmacological properties, contain bioactive compounds that offer anti-inflammatory, antioxidant, antimicrobial, and soothing effects. From ancient civilizations to contemporary medicine, medicinal plants have been harnessed to manage pain, inflammation, and skin disorders [2,3]. Their ability to address both physical symptoms and emotional well-being makes them invaluable in holistic healthcare.

In the context of palliative care, where the focus shifts from curative interventions to improving quality of life, the role of medicinal plants becomes increasingly significant [4]. Patients in palliative care often endure complex symptoms such as chronic pruritus, which can arise from systemic diseases, cancer treatments, or medication side effects [5]. Pruritus is more than just a physical discomfort—it can disrupt sleep, exacerbate stress, and negatively impact the emotional and psychological state of the patient [6]. Traditional treatments for pruritus, such as antihistamines, corticosteroids, or immunomodulators, can sometimes fall short, either in effectiveness or due to undesirable side effects [7,8]. These limitations underscore the need for alternative therapies that are effective but also safe and gentle for prolonged use.

Medicinal plants offer a promising complementary approach to symptom management in palliative care [9]. Their active compounds, including flavonoids, terpenoids, and phenolic acids, work through various mechanisms to reduce inflammation, modulate the immune response, and relieve itching. Plants like chamomile (*Matricaria chamomilla*), licorice (*Glycyrrhiza glabra*), and calendula (*Calendula officinalis*) have demonstrated the potential to alleviate pruritus through their soothing and anti-inflammatory properties [10,11]. Beyond their physical benefits, the use of these plants aligns with the principles of palliative care by addressing the emotional and psychological dimensions of patient well-being, often offering a sense of comfort and connection to natural, gentle remedies.

This review aims to explore the potential of medicinal plants in managing pruritus in palliative care settings. It examines the pharmacological profiles, mechanisms of action, and clinical applications of selected plants, highlighting their ability to enhance comfort and quality of life for individuals facing life-limiting conditions. By integrating traditional botanical knowledge with modern scientific evidence, medicinal plants continue to provide innovative solutions for symptom management, bridging the gap between ancient practices and contemporary healthcare. As the interest in complementary therapies grows, understanding the role of medicinal plants in palliative care can pave the way for more integrative and patient-centered approaches to healthcare.

## 2. Pruritus—An Overview

Pruritus, commonly called itching, is an unpleasant sensation that elicits the urge to scratch. It is a multifactorial symptom that can arise from various dermatological, systemic, neurological, and psychogenic conditions. Pruritus can be classified as acute, lasting less than six weeks, or chronic, persisting for six weeks or longer. Chronic pruritus, in particular, can significantly impair the quality of life, disrupting sleep, concentration, and emotional well-being [7,8].

### 2.1. Mechanism of Pruritus

Pruritus involves complex interactions between the nervous and immune systems. Sensory nerve fibers in the skin, particularly C-fibers, are stimulated by pruritogens, which induce itching. These pruritogens include histamine, proteases, cytokines, and neuropeptides. Once activated, nerve fibers transmit signals to the spinal cord and brain, leading to the perception of itching [12,13].

There are two primary pathways for pruritus: histamine-dependent and histamine-independent. Histamine-dependent itching, triggered by mast cell degranulation, is commonly associated with allergic reactions and urticaria. Histamine-independent pathways, which involve cytokines such as interleukin-31 (IL-31) and proteases, play a role in chronic and systemic diseases like atopic dermatitis and chronic kidney disease [8,14].

### 2.2. Clinical Manifestations

Pruritus can present as a localized or generalized symptom. The skin may appear typical or exhibit signs of excoriation, erythema, or lichenification due to chronic scratching. The underlying cause often determines the specific characteristics and associated symptoms [15,16].

### 2.3. Causes of Pruritus

Pruritus can arise from a variety of underlying conditions. Dermatological causes include atopic dermatitis, psoriasis, urticaria, and eczema, all of which directly affect the skin [15,16]. Systemic diseases can also lead to pruritus, with common examples being liver disorders such as cholestasis, chronic kidney disease, and hematologic conditions like polycythemia vera [17,18]. Neurological conditions also contribute to pruritus, with cases linked to multiple sclerosis, post-herpetic neuralgia, and brachioradial pruritus [19,20]. Additionally, psychogenic factors, including stress, anxiety, and psychodermatological disorders, can play a significant role in triggering or exacerbating the sensation of itching [21,22,23].

### 2.4. Management of Pruritus

Managing pruritus involves addressing the underlying cause while alleviating the symptoms to improve the patient’s quality of life. Modern therapeutic approaches combine pharmacological treatments with supportive skincare to relieve itching and its discomfort.

For histamine-mediated pruritus, such as that seen in urticaria or allergic reactions, antihistamines remain the first-line treatment. These drugs block histamine receptors, reducing the itching sensation. While effective for acute conditions, antihistamines are often less beneficial for chronic or histamine-independent pruritus [14,24].

In chronic or inflammatory pruritus cases, such as atopic dermatitis or psoriasis, topical corticosteroids are commonly used to reduce inflammation. When corticosteroids are not suitable for prolonged use due to side effects, calcineurin inhibitors like tacrolimus or pimecrolimus may be prescribed as steroid-sparing agents. These drugs help modulate immune responses without the risks associated with long-term steroid use [25,26].

Moisturizers and emollients are universally recommended as supportive therapy to restore the skin barrier, particularly in conditions like eczema or xerosis. They reduce transepidermal water loss and protect against external irritants, alleviating dryness-associated itching [27,28,29].

For systemic or neuropathic pruritus, newer therapies are emerging. Gabapentinoids such as gabapentin or pregabalin are effective in neuropathic itching by modulating nerve signals. In chronic kidney disease-associated pruritus conditions, kappa-opioid receptor agonists, such as nalfurafine, have shown promise. IL-31 receptor antagonists, such as nemolizumab, target specific cytokines implicated in chronic itching and are becoming a valuable tool for difficult-to-treat cases [30,31].

Non-pharmacological strategies also play a role. Phototherapy, particularly narrowband UVB, is effective in managing certain pruritus types, such as chronic kidney disease or atopic dermatitis [32,33].

Complementary approaches, including medicinal plants like chamomile, peppermint, and aloe vera, are gaining attention for their anti-inflammatory and soothing properties. These remedies appeal to patients seeking natural or integrative treatments, though more research is needed to establish standard dosages and efficacy.

Overall, treatment choice depends on the type, severity, and underlying cause of pruritus, often requiring a personalized approach to ensure effective symptom control and patient comfort.

## 3. Research Methods

This article focuses on using medicinal plants in various dermatological conditions characterized by pruritus. For this purpose, the search was performed in the following online databases: b-on (https://www.b-on.pt/ (accessed on 1 September 2024)), PubMed (https://pubmed.ncbi.nlm.nih.gov/ (accessed on 1 September 2024)), Web of Science (https://webofknowledge.com/ (accessed on 1 September 2024)), and Scopus (https://www.scopus.com/ (accessed on 1 September 2024)) for any in vitro, animal, or clinical studies investigating the efficacy of a medicinal plant or its bioactive component in managing of pruritus. The searched keywords were “plant”, “herb”, “extract”, “itch”, “pruritus”, “scratch”, and “antipruritic”. The title and abstract of each article were examined to eliminate duplicates. No limitation of a year range was used.

## 4. Antipruritic Medicinal Plants and Their Bioactive Components for Managing Pruritus

The medicinal plants studied in this review are arranged alphabetically according to their scientific name, followed by family name, parts used, growing area, related clinical trials, main bioactive components, probable mechanism of antipruritic effect, usage and dosage, and safety and side effects. The summary of medicinal plants reviewed is presented in Table 1. Figure 1 shows a diagram illustrating the mechanism of action of the medicinal plant and its active phytochemicals.

### 4.1. Aloe barbadensis

*Aloe barbadensis* (Figure 2), widely known as aloe vera, is a succulent plant in the Asphodelaceae family, traditionally used for its skin-soothing and healing properties [34]. Native to arid regions of the Arabian Peninsula, it has been cultivated worldwide, particularly in tropical and subtropical areas, due to its medicinal potential [35,36]. Aloe vera gel, the clear, mucilaginous substance within the leaves, has long been applied for skin ailments, including burns, wounds, and pruritus. The plant’s hydrating, anti-inflammatory, and antimicrobial properties have made it popular in dermatology, especially for managing skin irritation and inflammation [34,37].

#### 4.1.1. Active Compounds

The therapeutic effects of *Aloe barbadensis* are attributed to a complex mix of bioactive compounds within the gel. Polysaccharides, primarily acemannan, form the core component and are responsible for the gel’s hydrating and film-forming properties [38,39] (Figure 3). Acemannan not only aids in moisture retention but also exhibits wound-healing properties by stimulating fibroblast activity and collagen synthesis [38,40]. The gel contains phenolic compounds, including flavonoids and saponins, contributing antioxidant and antimicrobial effects [41]. Other active components include enzymes (such as bradykinase, which has anti-inflammatory effects), salicylic acid (which provides mild analgesic and anti-inflammatory properties), amino acids, and vitamins C and E, both of which further enhance the gel’s skin-soothing and protective properties [34,42,43]. This unique composition makes aloe vera gel particularly effective for managing pruritus, promoting healing, and providing long-lasting hydration to the skin.

#### 4.1.2. Mechanism of Action

The therapeutic properties of Aloe barbadensis, commonly known as aloe vera, are primarily attributed to its polysaccharides, particularly acemannan, and other bioactive compounds, including vitamins, minerals, enzymes, and amino acids. Acemannan, a polysaccharide found in aloe gel, stimulates skin fibroblasts and enhances collagen synthesis, promoting wound healing and soothing inflamed skin. Aloe vera also contains glycoproteins and growth factors that modulate inflammation by inhibiting the production of specific pro-inflammatory cytokines, such as interleukin-6 (IL-6) and tumor necrosis factor-alpha (TNF-α) [44,45]. Its salicylic acid content provides mild analgesic and anti-inflammatory effects, further contributing to its antipruritic properties [34,37]. Additionally, aloe vera gel’s high water content supports hydration, while the polysaccharides form a protective film over the skin, improving moisture retention and creating a barrier against environmental irritants [46,47].

#### 4.1.3. Clinical Evidence

Clinical studies on *Aloe barbadensis* have demonstrated its efficacy in managing skin conditions associated with pruritus, including psoriasis, atopic dermatitis, and xerosis [48,49,50,51]. Topical aloe vera has significantly reduced itching, erythema, and scaling in mild-to-moderate psoriasis patients [52,53]. In cases of atopic dermatitis, studies have documented aloe vera gel’s ability to relieve itching and improve skin integrity due to its anti-inflammatory and moisturizing effects [54,55]. For xerosis (dry skin), aloe vera enhanced skin hydration and reduced itching in patients with dry skin conditions [55,56]. While positive outcomes are often reported, further research with larger sample sizes and standardized formulations is needed to establish the consistency of aloe vera’s effects on pruritus.

#### 4.1.4. Usage and Dosage

*Aloe barbadensis* is primarily used topically in gel form for pruritus relief. The clear gel inside the leaf can be applied directly to the affected skin area up to three times a day [57]. Commercially prepared aloe vera creams and lotions are also available, typically containing 0.5% to 1% aloe extract, and are recommended for use once or twice daily, depending on skin sensitivity and the severity of symptoms [58,59]. For patients with sensitive skin, a patch test is advisable, as some individuals may experience mild irritation. Aloe vera can also be taken orally in capsule or juice form for systemic anti-inflammatory effects, but consultation with a healthcare provider is recommended for oral administration due to potential gastrointestinal side effects at higher doses [58].

#### 4.1.5. Safety and Side Effects

*Aloe barbadensis* is generally safe when used topically, with rare adverse reactions. Minor side effects may include mild skin irritation or itching, particularly in individuals with sensitive skin or when applied to broken skin [34,58,60]. Ingesting aloe vera gel or latex in large amounts can lead to gastrointestinal symptoms, such as cramps, diarrhea, and electrolyte imbalances, and it is not recommended for pregnant or lactating women due to insufficient safety data [58,61,62,63]. Aloe latex, which contains anthraquinones, has a strong laxative effect and should be avoided in oral preparations, as it may be toxic in high doses [58,61]. Aloe vera may also interact with certain medications, particularly blood thinners and anti-diabetic drugs [64]. When used within recommended guidelines, aloe vera is generally well-tolerated, providing safe and effective relief for pruritic conditions.

### 4.2. Calendula officinalis

*Calendula officinalis* (Figure 4) is a plant belonging to the Asteraceae family commonly known as marigold, field marigold, garden marigold or marigold [65]. Due to its long flowering period, its name derives from the Latin “Calend”, which means the first day of each month. It is called the “*herb of the sun*” because the flowers bloom in the morning, and the petals close at night [65,66]. It is an annual herbaceous plant cultivated throughout the temperate zone of the world, and it varies between 30 and 60 cm in height, with fasciculate roots, a short, solid, angular stem, which can be erect or prostrate, pubescent, slightly toothed, and have lanceolate and alternate leaves [65,67].

The first medicinal use of *C. officinalis* occurred in the Middle Ages when it was used to treat digestive problems, menstrual cramps, various types of skin lesions, liver obstructions, and snake bites, and to strengthen the heart. In the 18th century, flowers treated headaches, jaundice, and redness. During the Civil War, *C. officinalis* was used as a healing agent for wounds and injuries and as a medicine to treat measles, smallpox, and jaundice [66,67]. In traditional and homeopathic medicine, *C. officinalis* is used for vision problems, menstrual irregularities, varicose veins, hemorrhoids, and duodenal ulcers [68]. This plant has therapeutic purposes recognized by the European Medicines Agency since 2008, including anti-inflammatory, antispasmodic, healing, antioxidant, antibacterial, antifungal and immunostimulant properties, and has also been applied and recommended for the prevention and treatment of radiodermatitis [69]. Use in treating skin lesions is one of the most important [44], mainly because of its anti-inflammatory, immunostimulating, healing, antibacterial, and antifungal properties [65].

**Figure 4 plants-13-03515-f004:**
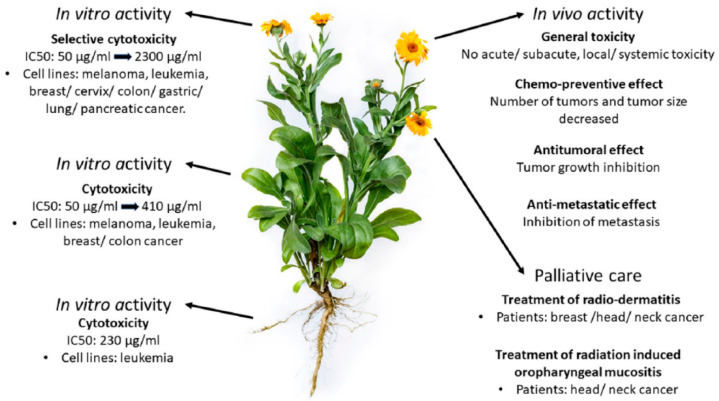
*Calendula officinalis*. Adapted from Cruceriu et al. (2018) [69].

#### 4.2.1. Active Compounds

In general, leaves, flowers, and roots of *C. officinalis* have been reported to have therapeutic potential [70,71]. In previous studies, numerous chemical compounds, such as flavonoids, phenolic acids, saponins, carotenoids, sterols, lipids, etc., are present in this species [72,73]. These compounds have been shown to exhibit a wide range of pharmacological effects related to anti-inflammatory, anti-HIV, anti-cancer, immunostimulating, and antibacterial and antioxidant activities [74,75,76]. According to Figure 5, the main constituents identified are phenols (phenolic acids, flavonoids), terpenes (monoterpenes, sesquiterpenes, saponins, carotenoids), and alkaloids [74,77].

Terpenoids, including carotenoids, polyphenols (phenolic acids and flavonoids), triterpenoids (steroids, which are its main naturally occurring group), and polysaccharides, are among the main active compounds in *C. officinalis* [78]. These compounds contribute to the plant’s medicinal qualities and broad use in the food, pharmaceutical, and cosmetic industries to improve skin tone and texture, promote protection against UV radiation, and support cellular functions [79]. Polyphenols, including flavonoids and phenolic acid, are some of the most important natural substances with biological characteristics. In particular, the phenols of *C. officinalis* have cardio-protective, anti-inflammatory, and antioxidant functions [80]. Triterpenes, sesquiterpenes, monoterpenes, and tetraterpenes (carotenoids) comprise most of the terpenic profile of *C. officinalis*. Sesquiterpenes have antioxidant [71] and antimicrobial functions [81], and the triterpenes have cytotoxic and anti-cancer [82], anti-diabetic and hypoglycemic [66], anti-inflammatory [83], wound-healing [84], hepato-protective [85], and nephron-protective properties [66]. Alkaloids, including sitsirikine, vinblastine, vindoline, catharanthine, and vinleurosine, have been identified in the flowers of *C. officinalis* [86]. Furthermore, pyrrolizidine alkaloids of the platynecine type have been found in aerial parts accounting for 41.5% of the total [87].

**Figure 5 plants-13-03515-f005:**
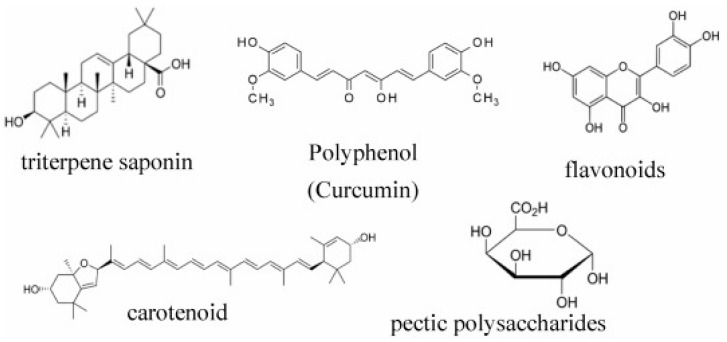
The molecular structure of some of the main active compounds in *C. officinalis* [88].

#### 4.2.2. Mechanism of Action

Many studies have established that *C. officinalis* has a broad spectrum of pharmacological effects, but most of its mechanisms of action are poorly understood. Polyphenols are believed to alter cellular and molecular signaling pathways to exert anti-oxidative, anti-inflammatory, and anti-cancer effects [89]. The antioxidant properties of *C. officinalis* occur along with its immune-modulating properties. It can contribute to the reduction in inflammation by inhibiting immune cell activity. *C. officinalis* also exhibits anti-inflammatory properties by preventing the synthesis of prostaglandins. When an injury occurs, prostaglandins are produced, causing pain and swelling. The anti-inflammatory effects of *C. officinalis* occur by inhibiting pro-inflammatory cytokines (IL-6, IL-1β, TNF-α, and IFN-γ, etc.), COX-2, prostaglandin synthesis, iNOS (inducible nitric oxide synthase), and CRP (C-reactive protein) [90].

*C. officinalis* was effective against Gram-positive (*Staphylococcus aureus* and *Clostridium perfringens*), Gram-negative (*Salmonella typhi* and *Pseudomonas aeruginosa*) [91], and MRSA bacteria [92]. Either alone or in conjunction with other bioactive molecules found in extracts, such bioactive components may prevent the growth of bacteria. Their capacity to interact with bacterial cell walls and other soluble and cellular proteins confers their antibacterial properties [91]. *C. officinalis* promotes wound healing through decreased inflammation, enhanced blood flow, and antibacterial action [93]. Calendula may enhance blood flow to the wound site to provide vital nutrients and oxygen for tissue regeneration [94]. Incorporating *C. officinalis* into dressings for treating wounds reduces the proliferation of *Escherichia coli* and *Staphylococcus aureus* [95]. Flavonoids, triterpenoids, and essential oils have all been implicated in the antibacterial properties of *C. officinalis*. Antibacterial effects have been attributed to several potential mechanisms, including cytoplasmic membrane damage, protein denaturation, enzyme system disruption, and decreased ATP generation [96,97,98]. Numerous studies have demonstrated calendula’s anti-cancer qualities [99,100]. This process involves caspase signaling to induce apoptosis in cancer [101].

An experimental rat model demonstrated that a therapeutic dose of *C. officinalis* dry extract improved the course of acute toxic hepatitis and lessened the severity of cytolysis and cholestasis, which are the primary signs of liver damage, and the energy processes in hepatocytes. Moreover, it decreased the intensity of free radical oxidation and limited the inflammatory response [102]. *C. officinalis* has also been shown to have the ability to protect the heart through calcium channel blockade [103].

##### Antipruritic Effect

Itching can be incapacitating and highly challenging to treat efficiently. Plants from the Asteraceae family are widely used as ethno medicines to treat pruritus [104], and they are the most prevalent family that alleviates itching [105]. Herbal treatment for pruritic disorders has been used for millennia, but additional research with a greater level of in-depth inquiry is required. No study has been found in the literature that validated the mechanism of action of *C. officinalis* in relieving pruritus.

Symptomatic treatment of itch may include treatment with topical emollients, corticosteroids, calcineurin inhibitors, and crisaborole. Other topical antipruritic agents include menthol, capsaicin, antihistamines, naltrexone, and N-palmitoylethanolamine [106]. However, *C. officinalis* oil is rich in fatty acids that help to rebuild a weakened skin barrier system [107] and protect against UV radiation [108]. The saponins in *C. officinalis* flowers have been shown to have high anti-inflammatory activity, enhancing the relief of the inflammatory process [90,109]. The anti-inflammatory properties of *C. officinalis* flos are linked to their flavonoid and triterpene derivative concentration. *C. officinalis* flower isorhamnetin 3-glycosides inhibited lipoxygenase. Oleanane-type triterpene glycosides exhibited marked anti-inflammatory activity against TPA-induced inflammation in the mouse ear [110]. When administered topically, *C. officinalis* flos extract—which primarily consists of triterpenoids—inhibited the edema caused by croton oil in vivo [111]. By soothing this inflammatory process, *C. officinalis* potentially relieves the characteristic symptoms of psoriasis [112,113] and dermatitis [88,114,115], such as redness, itching, and swelling.

#### 4.2.3. Clinical Evidence

Limited data have been published on the relief of itching by *C. officinalis,* as shown in Table 2.

**Table 2 plants-13-03515-t002:** Utilization of calendula in pruritus.

Type of Study/Pathology	Dosage of Calendula	Principal Outcomes	Reference
Antipruritic effects of the vaginal gel of *C. officinalis* in treating vaginal dystrophy	Vaginal gel containing aglicon isoflavones 10 mg, *L. sporogenes* 109 CFU, *Calendula officinalis* 125 mg, and lactic acid 20 mg (=Estromineral Gel, Rottapharm-Madaus)	The severity of itching, burning, vulvovaginal erythema, vaginal dryness, and dyspareunia significantly reduced during vaginal gel treatment compared with the no topical treatment group.	[116]
Diabetic feet and injuries with signs of infection and itching	*C. officinalis* cream was applied twice daily.	The infection process blocked, reducing itching, redness, pain, dryness’ Disappearance of various scars.	[117]
Effects of *C. officinalis* cream on bacterial vaginosis and pruritus	*C. officinalis* extract-based cream administered intravaginal for 1 week	Itching was significantly more common in the *C. officinalis* group than in the metronidazole group.	[118]
Oil Mixtures for treating Xerosis with Pruritus in Elderly People	Oil mixtures (*C. officinalis*, Lavender, Chamomile, Rosemary) on the volar legs twice a day for 4 weeks.	Pruritus severity of the oil mixtures group was significantly decreased than Virgin Coconut Oil (VCO) Skin hydration and sebum levels of the plants’ oil mixture group were significantly increased compared to VCO. Clinical dry score, pruritus severity, skin hydration, and skin sebum levels of plant oil mixtures were significantly better than VCO.	[119]
Topical *C. officinalis* on prevalence of radiation-induced dermatitis (pruritus)	Topical *C. officinalis* (≤5% *v*/*v*) 2–3 days before radiation therapy	Randomized trial showed no difference between *C. officinalis* and standard of care (Sorbolene) for preventing dermatitis and pruritus.	[120]

#### 4.2.4. Usage and Dosage

Studies have reported using *C. officinalis* in topical formulations (cream or oil). There is a nurse recommendation for using *C. officinalis* cream before skin cleansing. In diabetic feet and injuries, the cream was applied with a gentle massage until fully absorbed, without covering the injuries [117]. To prevent radiation-induced dermatitis, topical *C. officinalis* (≤5%)/2–3 days before the commencement of radiation therapy was recommended [120]. An ointment containing 1.5% of the total extract obtained from *C. officinalis* flowers (3 times a day for 10 days) has been established as a safe and effective treatment for infant diaper dermatitis [121]. During xerosis treatment, it is recommended that the cream be applied twice a day [119]. Additionally, *C. officinalis* hydrolate (INCI: *Calendula officinalis* L. flower extract) can be used as a gentle skin toner or spray for soothing irritation, especially for sensitive or inflamed skin [83,122].

#### 4.2.5. Safety and Side Effects

There are no known contraindications or side effects associated with *C. officinalis* use. The plant’s allergy risk is shallow because it contains no henalin (sesquiterpene lactones) [123]. *C. officinalis* is often used for wound care and has been approved by the German Health Commission for healing wounds and leg ulcers. The European Medical Agency’s (EMA) Committee on Herbal Medicinal Products (HMPC) recommends the use of *C. officinalis* flower medications for mouth or throat discomfort, minor wounds, and skin inflammation [124]. There are also indications of calendula’s usage as herbal medicine in British Herbal Pharmacopoeia. These include acute and chronic inflammatory skin lesions, duodenal ulcers, sebaceous cysts, and enlarged or inflamed lymphatic nodes [125]. Calendula is best administered orally or topically under EMA monographs and the European Scientific Cooperative on Phytotherapy (ESCOP).

Regarding effectiveness and safety, *C. officinalis* has considerable potential for developing innovative medications to treat various human ailments [70]. To fully realize calendula’s potential, clinical studies examining its safety and effectiveness—especially for youngsters and expectant mothers—are essential [88].

### 4.3. Curcuma longa

Turmeric, *Curcuma longa* L. (Figure 6), belongs to the ginger family Zingiberaceae, a perennial plant moderately tall, with underground rhizomes that are mostly ovate, pyriform, oblong, and often short-branched [126]. The most common synonyms are *Curcuma domestica* Vale, *Curcuma domestica* Loir, *Amomum curcuma* Jacq, Curcuma, yellow root, sun root, Indian saffron, and yellow ginger. It is cultivated in tropical and subtropical regions like Pakistan, Nepal, Thailand, Cambodia, China, Japan, Indonesia, Malaysia, Madagascar, Philippines, Peru, Jamaica, and Haiti but is native to India, Bangladesh, and Sri Lanka [127].

Turmeric’s use dates back over 4000 years to the Vedic age in India. Throughout history, turmeric has played a significant role in folk medicine, particularly in Ayurveda and traditional Chinese medicine, where it has been used to treat a wide range of ailments, from simple to chronic illnesses. It was introduced to Europe in the 13th century by Arab traders, but it remains less commonly used there compared to its prevalent use in Eastern countries [129]. Even so, it is one of the most popular medicinal herbs with broad applicability, including antioxidant [130], anti-inflammatory [131], antimutagenic [132], antimicrobial [133], and anti-cancer [134], among others. Turmeric is a rich source of bioactive compounds such as antioxidants, polyphenols, and flavonoids, which can serve as alternatives to antibiotics in food and food products [135].

Additionally, these compounds have potential applications in treating various conditions, including skin diseases. The skin is the largest organ in the human body and is fundamental in protecting against continuous environmental aggressions. They can lead to several disorders like chronic pruritus, psoriasis, atopic dermatitis, and other inflammatory conditions. Several studies have shown that formulations containing curcumin exhibited significant improvement in reducing itch intensity due to the ability to inhibit inflammatory pathways and reduce oxidative stress [136,137,138,139,140,141,142].

It has been used through different routes of administration, topically on the skin for wounds, itch, blistering diseases such as pemphigus and herpes zoster, parasitic skin infections, and acne. It is administered orally for the common cold, liver, and urinary tract diseases and used as a blood purifier. For chronic rhinitis and coryza, it has been used via inhalation [143].

Natural products have been used in traditional medicines for thousands of years and have shown promise as a source of components for developing new drugs [144]. There is rising interest within the conventional medical community in discovering new, affordable, and safe molecules for treating inflammatory and neoplastic diseases. Growing evidence indicates that curcumin could be an effective agent in treating various skin conditions [145].

#### 4.3.1. Active Compounds

The rhizomes contain curcumins, the yellow pigment, and its powder is used extensively as a coloring and flavoring agent, especially in curries and mustards, but also has spice and food preservation [146]. It is the most studied active turmeric compound because of its properties and the pharmacological activity attributed to curcuminoids. The principal is curcumin (1,7-bis(4-hydroxy-3-methoxyphenyl)-1,6-heptadiene-3,5-dione), also called diferuloylmethane, a member of the diarylheptanoid class of natural products, is a phenolic phytochemical extracted from the *Curcuma longa* rhizome [147]. The other major curcuminoids present in turmeric are demethoxycurcumin, bisdemethoxycurcumin, and cyclocurcumin. Together, they are termed the curcuminoid complex [148].

The volatile oils are turmerone (anti-inflammatory and antimicrobial properties), atlante (potential neuroprotective effects), and zingiberene (active in anti-inflammatory and antioxidant properties of turmeric). Turmeric also has polysaccharides, which are associated with immunomodulatory effects, the protein turmerin, which has antioxidant properties, and some vitamins (such as vitamins C and B), minerals (such as manganese, iron, and potassium), and essential oils, which contribute to aroma and additional therapeutic effects [149]. Figure 7 shows the major active compounds in turmeric.

#### 4.3.2. Mechanisms of Action

Like nonsteroidal anti-inflammatory drugs, curcumin may exert its effects through a single mechanism or a combination of mechanisms, such as inhibiting arachidonic acid metabolism, cyclooxygenase (COX), prostaglandin (PG) synthesis, lipoxygenase (LOX), and cytokines interleukin (IL) and tumor necrosis factor (TNF). It also inhibits the release of steroidal hormones from the adrenal glands and stabilizes lysosomal membranes [147,151,152]. Clinical studies have additionally validated curcumin’s efficacy in reducing inflammation in patients post-surgery [153].

Curcumin has gained global recognition for its numerous health benefits, and these benefits, according to some authors, are most effectively realized when curcumin is paired with compounds like piperine [127,147,154], which substantially enhances its bioavailability. It has demonstrated potent antioxidant activity, effectively scavenging reactive oxygen species (ROS) and inhibiting lipid peroxidation. However, the capacity of curcumin to reduce and scavenge ROS in the highly polar environment of the cytoplasm remains unclear due to its highly lipophilic nature [155].

The role of curcuminoids in a high spectrum of diseases is well described and is related to central nervous system diseases [156,157], respiratory diseases [158,159], cardiovascular protection [160,161], gastrointestinal diseases [162,163], liver diseases [164,165], metabolic disorders [166,167], cancer treatment [168,169], and genitourinary tract diseases [170,171].

Another subject that has been increasing the interest of many authors is skin diseases. The most abundant studies are about psoriasis [172,173], atopic dermatitis [139,145], iatrogenic dermatitis [77,174], anti-aging [175,176], wound healing [177,178], acne [137,179], and skin infections [138,141]. Since curcumin induces apoptosis of inflammatory cells, it may expedite the healing process by shortening the inflammatory phase. Additionally, curcumin could enhance collagen synthesis and promote the migration and differentiation of fibroblasts [180].

##### Antipruritic Effect

Pruritus is commonly associated with an irritating sensation that evokes the desire to scratch, severely decreasing the quality of life of affected individuals. Depending on the cause of itchiness, generally associated with systematic diseases, medication, and external aggressions, the skin may appear normal or exhibit signs of inflammation, roughness, or bumps. Continuous scratching can lead to thickened, raised areas of skin that may bleed or become infected [137]. The most severe diseases associated with pruritus are uremic and chronic pruritus.

In general, itching can be classified as histamine-dependent or histamine-independent. It acts on the C-nerve fibers that perceive pain and itching, and the stimulus is transmitted to the brain as itching and to the nerve endings, causing the release of neuropeptides neurotransmitters [136]. Mast cells function as a “powerhouse”, releasing allogenic and pruritogenic mediators such as proteoglycans, proteases, leukotrienes, biogenic amines, cytokines, and chemokines. These mediators stimulate reciprocal interactions with specific sensory nerve fibers [181].

It has been shown that IL-31 induces acute itch by acting on IL-31RA in sensory nerves, while IL-4 and IL-13 induce chronic itch by acting on sensory nerves and lowering the response threshold to pruritogenic factors [182].

The antipruritic mechanism of curcumin is not explored; however, it is known that it has antipruritic effects that inhibit cytokine expressions of TNF-α and IL-4 and the cutaneous anaphylaxis reaction induced by the IgE-antigen complex and the scratching behavior induced by compound 48/80 [183].

#### 4.3.3. Clinical Evidence

Not many publications relate to the utilization of turmeric in pruritus; most studies consider it a side effect of other skin disorders. Table 3 shows the utilization of turmeric in pruritus.

#### 4.3.4. Usage and Dosage

The primary issue with ingesting curcumin is its low bioavailability, mainly due to poor absorption, rapid metabolism, and quick elimination. To enhance curcumin’s bioavailability, various agents that work primarily by blocking the metabolic pathway of curcumin have been tested to target these mechanisms. For instance, piperine, the main active component of black pepper and a known bioavailability enhancer, increases curcumin’s bioavailability by 2000%. Thus, adding agents like piperine can effectively resolve the problem of low bioavailability, resulting in a more effective curcumin complex [187].

#### 4.3.5. Safety and Side Effects

Curcuminoids have been classified as “Generally Recognized as Safe” (GRAS) by the US Food and Drug Administration (FDA) [188]. Clinical trials have demonstrated good tolerability and safety profiles, even at doses ranging from 4000 to 8000 mg/day [189] and with doses up to 12,000 mg/day [190] of a 95% concentration of three curcuminoids: curcumin, bisdemethoxycurcumin, and demethoxycurcumin. The results of some studies suggested that eventual toxicity does not appear to be dose-related.

Curcumin has shown excellent tolerance at oral doses. Since achieving systemic bioavailability of curcumin or its metabolites may not be crucial for resolving pathologies, it may have action both at a preventive and complementary level. Further research is justified to explore its potential as a long-term agent.

### 4.4. Glycyrrhiza glabra

*Glycyrrhiza glabra* (Figure 8), commonly known as licorice, is a perennial herb native to southern Europe, Asia, and parts of the Middle East [191]. It has been used for centuries in traditional medicine due to its potent anti-inflammatory, antimicrobial, and soothing properties [177,191,192]. The primary medicinal part of licorice is its root, which contains active compounds such as glycyrrhizin and flavonoids [193,194]. These bioactive substances are well-regarded for their therapeutic effects on skin conditions, especially those involving inflammation and pruritus [192,195]. Licorice is available in several forms, including extracts, hydrolates, and powders, making it a versatile component in creams, ointments, and hydrolates designed to alleviate symptoms of eczema, dermatitis, and allergic reactions [29,196].

#### 4.4.1. Active Compounds

The medicinal benefits of *Glycyrrhiza glabra* stem from its bioactive compounds, primarily glycyrrhizin, glycyrrhetinic acid, liquiritin, liquiritigenin, and isoliquiritigenin [193] (Figure 9). Glycyrrhizin is known for its potent anti-inflammatory and immunomodulatory effects, often utilized in managing inflammatory skin conditions [198]. Glycyrrhizin exerts effects by inhibiting the enzymes involved in inflammation, including COX and phospholipase A2, and reducing free radical activity, which is crucial for calming irritated skin [191,199]. Glycyrrhetinic acid, derived from glycyrrhizic acid, plays a role in anti-inflammatory and antioxidant pathways. It has downregulated pro-inflammatory cytokines, such as TNF-α and IL-6, commonly elevated in pruritic skin conditions [200]. Liquiritin and liquiritigenin are flavonoids with potent antioxidant properties. They help neutralize ROS, thereby reducing oxidative stress in the skin and preventing further irritation that can exacerbate itchiness [201]. Liquiritin also supports skin hydration and may aid in pigment regulation, promoting skin recovery [202]. Isoliquiritigenin, another flavonoid, demonstrates anti-inflammatory, antioxidant, and antimicrobial effects. It inhibits histamine release from mast cells, a critical aspect of managing allergic itch responses [203].

#### 4.4.2. Mechanism of Action

The antipruritic effects of *Glycyrrhiza glabra* are primarily attributed to its active compounds, especially glycyrrhizin, which has a range of anti-inflammatory and immune-modulating properties [198]. Glycyrrhizin, the primary bioactive compound in licorice, inhibits enzymes involved in inflammation, including COX and phospholipase A2 [205]. This inhibition reduces the production of pro-inflammatory mediators like prostaglandins and leukotrienes, commonly elevated in skin conditions associated with pruritus [206]. By lowering inflammation, licorice helps reduce the sensitivity of nerve endings responsible for itch sensation. Glycyrrhizin and other flavonoids in licorice downregulate pro-inflammatory cytokines such as TNF-α and IL-6, which are significant in inflammatory responses [198,207]. This cytokine modulation reduces immune cell activation in the skin, leading to decreased inflammation and relief from itchiness, especially in conditions like eczema and dermatitis. Flavonoids in licorice, including glabridin and liquiritin, possess potent antioxidant properties, which help neutralize ROS that contribute to skin irritation and inflammation [208]. By reducing oxidative stress, these antioxidants protect skin cells from further damage and prevent the exacerbation of itchiness. Studies suggest that licorice compounds may inhibit histamine release from mast cells, a critical mediator of allergic reactions that commonly cause itching [24,209]. By reducing histamine release, licorice helps manage allergic itch responses, making it beneficial for treating itch associated with allergies.

##### Antipruritic Effect

Through these mechanisms, licorice reduces the inflammatory response and the itch sensation at the nerve level, effectively soothing pruritic symptoms. This multi-faceted approach makes licorice valuable in creams, ointments, and hydrolates for managing pruritus, particularly in inflammatory and allergic skin conditions like eczema, psoriasis, and atopic dermatitis.

#### 4.4.3. Clinical Evidence

Several studies have investigated the antipruritic properties of licorice in various skin conditions characterized by itching [199]. These studies emphasize the role of licorice’s active compounds, such as glycyrrhizin and glycyrrhetinic acid, in alleviating pruritic symptoms [200]. Clinical trials have demonstrated that licorice-based creams containing 1–2% licorice extract significantly reduce itching and erythema in patients with eczema and atopic dermatitis [195,196]. Glycyrrhizin’s anti-inflammatory properties, achieved through inhibition of pro-inflammatory cytokines like TNF-α and IL-6, were identified as key mechanisms for reducing symptoms [192,210]. Applying licorice hydrolate as a topical spray showed improved skin hydration and reduced pruritus in sensitive skin conditions. These effects are attributed to the flavonoid liquiritin, which enhances skin hydration and reduces oxidative stress by scavenging ROS [204,211]. In a study focused on histamine-induced pruritus, topical licorice formulations effectively inhibited histamine release from mast cells. This mechanism directly reduces itch severity in allergic conditions, making licorice particularly effective for managing allergic dermatitis [212]. Glycyrrhizin-containing ointments were reported to alleviate pruritus associated with psoriasis by reducing prostaglandin and leukotriene production. This finding was corroborated by studies examining the role of COX inhibition in the therapeutic action of licorice [29,199]. Licorice-based creams are compared favorably with low-dose corticosteroid ointments for managing mild-to-moderate pruritus. Although corticosteroids had faster initial effects, licorice provided sustained relief with fewer side effects, making it a viable alternative for long-term management [106,213].

#### 4.4.4. Usage and Dosage

For topical applications, a 1–2% concentration of licorice extract is recommended in creams or gels, applied to affected areas twice daily [214]. Alternatively, *G. glabra* hydrolate (INCI: Glycyrrhiza Glabra Water) can be used as a mild, soothing spray for sensitive skin. This hydrolate form provides gentle antipruritic effects and is particularly suited for sensitive or damaged skin that may react to essential oils.

#### 4.4.5. Safety and Side Effects

Licorice is generally safe when applied topically [197]. However, it should be used in moderation, as high concentrations of glycyrrhizin may cause skin irritation in sensitive individuals. For those with sensitive skin, conducting a patch test before application is advisable. *Glycyrrhizin* can affect blood pressure when absorbed in large amounts; however, this is unlikely with typical topical use [215,216].

### 4.5. Matricaria chamomilla

*Matricaria chamomilla*, commonly known as chamomile (Figure 10), is a medicinal herb widely used in traditional medicine for its anti-inflammatory, soothing, and antipruritic properties [217]. Native to Europe and Asia, chamomile has been recognized for its therapeutic effects in managing skin disorders such as eczema, atopic dermatitis, and psoriasis [218]. The plant’s flowers are rich in bioactive compounds that contribute to its pharmacological activities, including relief from pruritus.

#### 4.5.1. Active Compounds

Chamomile’s pharmacological activity is attributed to its rich profile of bioactive compounds, primarily concentrated in its flowers (Figure 11). Chamazulene, a notable component, exhibits potent anti-inflammatory and skin-soothing effects, effectively calming irritated skin [218]. Apigenin, a flavonoid in chamomile, is recognized for its powerful antioxidant and anti-inflammatory properties, contributing to its therapeutic benefits [220,221]. Bisabolol, another critical compound, enhances skin repair processes while promoting anti-inflammatory responses, further supporting its role in alleviating pruritus and maintaining skin health [222,223].

#### 4.5.2. Mechanism of Action

The antipruritic effects of *Matricaria chamomilla* are mediated through multiple pathways. Its bioactive components exhibit significant anti-inflammatory activity by inhibiting the production of pro-inflammatory cytokines, such as interleukin-6 (IL-6) and tumor necrosis factor-alpha (TNF-α), effectively reducing inflammation and irritation [218,225]. Additionally, chamomile promotes skin healing and enhances hydration by supporting the repair of the skin barrier and improving moisture retention, which is particularly beneficial for pruritus associated with xerosis or inflamed skin [220]. Furthermore, chamomile’s compounds modulate neurological activity by interacting with sensory nerve endings, thereby mitigating the transmission of itch signals and relieving pruritus [48].

#### 4.5.3. Clinical Evidence

Chamomile has shown considerable promise in managing pruritus through its application in various formulations, with both traditional and modern methods supporting its therapeutic benefits. Compresses made with cooled chamomile tea have been particularly effective for localized itching [226,227]. When applied topically, these compresses provide immediate relief by calming inflamed areas and soothing irritation, making them a practical remedy for acute itching episodes.

The use of chamomile essential oil, when appropriately diluted, has demonstrated efficacy in alleviating chronic pruritic conditions such as atopic dermatitis [228,229]. Its anti-inflammatory and antipruritic properties target underlying inflammatory processes while addressing the itch–scratch cycle exacerbating skin conditions. This formulation is particularly beneficial for patients seeking alternatives to conventional treatments like corticosteroids, as it offers relief with a lower risk of side effects.

Chamomile hydrolates (INCI: *Matricaria recuitita* (Chamomile) Water) and chamomile-infused baths provide an effective solution for managing widespread pruritus, especially in cases where large areas of the skin are affected [220,230]. The hydrolates, being gentle and water-based, are well-suited for sensitive or compromised skin, delivering soothing relief without the risk of irritation. Chamomile-infused baths allow full-body exposure to its therapeutic effects, making them an excellent option for conditions such as atopic dermatitis, eczema, or xerosis, where generalized itching is a significant concern.

Although these applications have shown promising outcomes, the evidence primarily stems from smaller studies and anecdotal use. Large-scale, controlled clinical trials are urgently needed to validate these findings, determine standardized dosages, and assess the long-term safety profile of chamomile-based treatments. This will ensure a more robust understanding of chamomile’s role in pruritus management and facilitate its integration into evidence-based medical practices.

#### 4.5.4. Usage and Dosage

Chamomile is administered in several forms to address pruritus. Topical application is the most common method for managing pruritus and skin irritation. Chamomile can be prepared as a hydrolate, compress, or cream, depending on the specific condition and severity of symptoms. Hydrolates, derived through steam distillation of chamomile flowers, are applied directly to the skin as sprays or toners for gentle and soothing relief. Chamomile tea is prepared for compresses by steeping the dried flowers in hot water, cooling the infusion, and applying it with a soft cloth to the affected areas. This method is particularly effective for localized pruritus and minor skin irritations. Chamomile extracts are also incorporated into topical creams or ointments, typically at 1% to 5% concentrations, and applied to the skin one to three times daily [218,220]. The duration of use varies depending on the condition being treated, with short-term application often sufficient for mild irritation and longer-term use recommended for chronic skin conditions such as eczema.

#### 4.5.5. Safety and Side Effects

*M. chamomilla* is generally well-tolerated; however, individuals allergic to plants in the Asteraceae family may experience adverse reactions [231]. A patch test is recommended before topical use. While rare, excessive use of concentrated preparations may lead to skin irritation.

### 4.6. Lavandula angustifolia

Lavender (*Lavandula angustifolia*) (Figure 12) is a medicinal plant widely recognized for its therapeutic and aromatic properties [232]. Used for centuries in folk medicine in different cultures, lavender has a broad spectrum of applications, including the treatment of anxiety disorders, insomnia, muscle pain, and dermatological conditions [50,233,234]. One of the most promising and recently studied applications is its use in treating pruritus, a common symptom in skin diseases such as eczema, psoriasis, atopic dermatitis, and urticaria [50,235]. Itching can be highly uncomfortable and, in chronic cases, lead to skin lesions, secondary infections, and a negative impact on the patient’s quality of life [236].

Itching can be triggered by various factors, including inflammation, skin barrier dysfunction, and altered immune response [237]. Lavender’s effectiveness in relieving pruritus is mainly related to its anti-inflammatory, soothing, and skin-regenerating properties, which derive from bioactive compounds present in the plant’s essential oil [232].

**Figure 12 plants-13-03515-f012:**
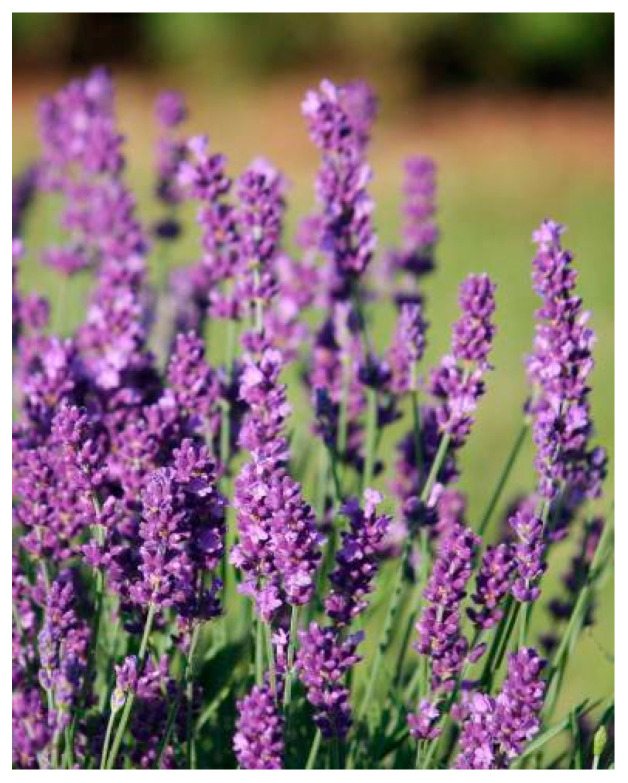
*Lavandula angustifolia*. Adapted from Jardim Botânico da UTAD [238].

#### 4.6.1. Active Compounds

*Lavandula angustifolia* contains several bioactive compounds. The most relevant for treating pruritus are those found in its essential oil (Figure 13). Among the main compounds are linalool and linalyl acetate, which are the major components, accounting for around 70% of lavender essential oil. In addition to these, other compounds such as camphene, beta-caryophyllene, terpineol, and 1,8-cineol also contribute to its medicinal properties [232,239]. Linalool is a monoterpene alcohol widely recognized for its calming and relaxing properties. In addition, linalool has an anti-inflammatory and analgesic action, directly reducing the itching and discomfort associated with pruritus [240]. Linalyl acetate, a monoterpene ester, is one of the main ingredients responsible for the soothing properties of lavender oil. Linalyl acetate acts as a natural anti-inflammatory and effectively regenerates damaged skin. It is also known for reducing skin sensitivity [241]. 1,8-Cineole is a compound with antiseptic and anti-inflammatory actions and is effective in treating secondary infections that can arise from scratching, itchy skin [242]. β-caryophyllene, a sesquiterpene that has a remarkable anti-inflammatory action and is known to interact with cannabinoid receptors in the body, which can help modulate the response to pain and inflammation, thus contributing to the relief of itching [243,244].

#### 4.6.2. Mechanism of Action

Lavender exerts its antipruritic properties through various biochemical and physiological mechanisms. The main mechanism of action is associated with the ability of its active compounds to inhibit the production of pro-inflammatory cytokines, such as interleukin-6 (IL-6) and tumor necrosis factor-alpha (TNF-α), both of which are responsible for the inflammatory response that contributes to the development of pruritus [246,247,248].

Another vital mechanism involves interaction with the nervous system. Linalool and linalyl acetate are known to modulate serotonergic and opioid receptors in the body, which play central roles in regulating the sensation of pain and itching. By acting on these receptors, lavender oil can reduce the hyperstimulation of sensory nerve fibers often associated with itching [249,250]. In addition, lavender’s antiseptic action can help prevent secondary infections from scratching the skin, which is common in cases of intense itching. This effect is beneficial in conditions such as eczema and atopic dermatitis, where open skin lesions can be a gateway for bacteria and other pathogens [251].

#### 4.6.3. Clinical Evidence

Clinical studies carried out in recent years have investigated lavender’s efficacy in treating dermatological conditions associated with pruritus. Although research is still limited, preliminary results are promising.

A randomized controlled study by Ro et al. (2002) [252] examined the effect of lavender essential oil on pruritus in patients undergoing hemodialysis. Patients who received topical applications of lavender oil reported a significant reduction in pruritus intensity compared to the control group after three weeks of treatment. The study suggested that lavender oil may be a safe and effective therapy for patients with chronic pruritus related to kidney failure.

A study by Dale and Cornwell (1994) investigated the effect of lavender oil on gestational pruritus, a common condition in pregnant women. The results showed that participants who used lavender essential oil diluted in a carrier oil significantly reduced pruritus symptoms with no significant adverse effects, indicating that lavender may be a safe option for pregnant women [253]. Silva et al. (2015) conducted a study to evaluate lavender essential oil’s antioxidant, anti-inflammatory, and antinociceptive effects. The results of this study revealed (in vivo) the analgesic and anti-inflammatory activities of lavender essential oil and demonstrated its significant therapeutic potential [254].

Hajhashemi et al. (2003) [239] argue that extracts are obtained from the leaves of *Lavandula angustifolia* Mill. (Lamiaceae) are used in Iranian folk medicine as remedies for treating various inflammatory diseases. To evaluate its probable analgesic and anti-inflammatory effects, they conducted an investigation whose results confirmed the traditional use of *Lavandula angustifolia* to treat painful and inflammatory conditions. Panahi (2024) investigated the effectiveness of a combination of herbal drops (Lamigex) composed of essential oils of *Syzygium aromaticum*, *Lavandula angustifolia*, and *Geranium robertianum* in relieving the symptoms of acute otitis externa and compared their effects with those of a 0.3% ciprofloxacin drop. The combination of herbal drops investigated proved effective in reducing the burden of infection and the symptoms of acute otitis externa [255].

A study was carried out by Abedian et al. (2020) [256] to investigate the effect of lavender on the pain and healing of episiotomy wounds. The systematic review found that using lavender (in any form) in the postpartum period significantly affects pain relief and episiotomy wound healing. Sheikhan (2012), in his research, also proved that the use of lavender oil essence can effectively reduce perineal discomfort after episiotomy [257]. Other clinical studies indicate that lavender oil positively impacts the correct acceleration of wound healing based on its antimicrobial and anti-inflammatory effects [258]. Jones (2011) [259] wrote an article examining the literature describing the efficacy of lavender oil in treating wounds to put these results into the context of perineal trauma. Simaei (2024) [260] carried out a study that aimed to assess the impact of lavender and metformin on patients with polycystic ovary syndrome (PCOS). The results showed that treatment with lavender and metformin significantly increased progesterone levels in PCOS patients. Lavender effectively increased progesterone levels and regulated menstrual cycles in PCOS patients, like metformin. Lavender could, therefore, be a promising candidate for the treatment of PCOS.

Another study explored the use of lavender in patients with atopic dermatitis. Treatment with lavender oil was associated with an improvement in the skin’s barrier function and a reduction in inflammation, leading to a noticeable decrease in the sensation of itching in patients with this chronic condition [235]. Kim et al. (1999) [261] studied the effects of lavender oil on mast cell-mediated immediate-type allergic reactions in mice and rats. The results indicated that lavender oil inhibits immediate-type allergic reactions by inhibiting mast cell degranulation in vivo and in vitro.

#### 4.6.4. Usage and Dosage

Lavender essential oil (INCI: *Lavandula angustifolia* Oil (Lavender)) can be used topically, diluted in carrier oils (such as coconut or almond oil), at a concentration of 2–5% [261]. To relieve itching, it is recommended to apply the mixture directly to the affected areas up to three times a day [262]. Another option is to use lavender compresses, made with 5 to 10 drops of the essential oil diluted in warm water [263]. The oil can also be incorporated into creams, lotions, or baths. Adding 10 to 15 drops of lavender essential oil to a bath can effectively treat generalized itching. This approach is beneficial in conditions such as atopic dermatitis, where large areas of the skin are affected. For a milder application, lavender hydrolate (INCI: *Lavandula angustifolia* (Lavender) Flower Water) can be used as a spray or toner on itchy or sensitive skin, providing soothing relief without dilution. This gentle approach is suitable for daily use and can complement other treatments, especially in sensitive or allergy-prone individuals [235].

In aromatic use, inhaling lavender oil can complement topical treatment, providing relaxation and stress relief, factors that often exacerbate pruritus [234]. Lavender can be used to treat itching in a variety of ways, with topical application being the most common and effective.

#### 4.6.5. Safety and Side Effects

*Lavandula angustifolia* is generally considered safe when applied topically in appropriate concentrations. Lavender essential oil should always be diluted before applying to the skin to avoid possible irritation [264]. People with sensitive skin or an allergy predisposition may experience contact dermatitis when using lavender-based products. In addition, lavender can cause photosensitivity reactions in rare cases, which means that the treated skin may become more sensitive to sunlight [265].

Pregnant women, nursing mothers, and young children should use lavender with caution, and it is recommended that they consult a health professional before using it. Inhalation of the essential oil should be avoided by people with respiratory problems, such as asthma, as it can potentially trigger asthmatic attacks [266].

### 4.7. Mentha piperita

Peppermint, *Mentha* × *piperita* L., belongs to the mint family [267] (Labiaceae or Labiatae botanical family) [268,269,270,271], a native herb of Mediterranean Europe that is cultivated in different parts of the globe (however, the USA is the largest producer worldwide, with around 4000 tons per year), and it is a plant that can be used in diverse forms, for example, in essential oil, leaf, leaf extract, and leaf water/hydrolate (Figure 14) [268,269,270]. As a plant, it is a cultivated triple hybrid believed to derive from *Mentha longifolia* L., *M. suaveolens* Ehr., and *M. aquatica* L. and, therefore, reproduced only vegetatively [269,270]. Other common names for this medicinal herb are mint-of-water-of-scent, spicy-mint, and mint-of-ladies [270]. Peppermint essential oil is a hybrid of spearmint (*Mentha spicata*) and water mint (*Mentha aquatica*) [268], which was discovered in 1696 in Mitcham (England), and the most common synonyms are *Mentha* oil, *M. piperita* oil, and oil of peppermint. It is recommended to use in moderation, in the correct dose, and for controlled periods to avoid neurotoxic effects 246,250,253]. It is a robust and sterile hybrid, reproducing by runners (i.e., stems that create roots in contact with the earth) and propagating quickly in fresh, uncompacted clay-limestone and humid soil. Its branches are quadrangular in shape; the leaves (opposite, oval, and pointed) have a violet-green color; the flowers have a pinkish color; the limbus is slightly arched; its ends are serrated, and the inflorescences are small spikes tight and short [271].

In the past, the mint liqueur was drunk in the morning to restore the body (liver) and the spirit (mind). After overeating and, above all, after drinking too much (anti-hangover action), its use is essential for digestive disorders (liver, stomach, and intestines—including irritable bowel syndrome) [269,270,271]. Even so, it is one of the most popular medicinal herbs with broad applicability, including antiemetic action (nausea and vomiting) [267,270,271,272], digestive tonic (stimulates digestion and relieves fullness) [270,271], the antispasmodic effect [267,269,270,271], and applicable in case of migraines and headaches [270,271,273,274]. Also, it is a heart tonic [271], antiseptic [270], analgesic [267,270,271], anti-inflammatory [271], and promotes relief from respiratory and skin ailments/diseases [269,270,271], among others. In the psycho-aromatherapy approach, peppermint is a stimulant, par excellence, of the central nervous system, increasing attention [273,275], learning, mood, etc., being neurotonic, combating mental fatigue, and dispelling distractions. Its fresh and cold aroma brings a positive attitude to life. It is calming for explosive tempers, anger, and wrath, as well as for that fear that paralyzes and stiffens the muscles [275]. Recently, peppermint’s role in managing pruritus (itchy skin) has gained attention, with studies examining its antipruritic properties attributed to menthol and other active compounds [267,268,276]. Peppermint leaves are a rich source of bioactive compounds such as essential oil, whose main constituents are the terpene alcohol menthol (30 to 50%), the terpene ketone menthone (10 to 60%) and, to a lesser extent, terpenic oxides (cineole), terpenes (e.g., limonene), and others [264,269,270,271]; free flavonoids (strongly methoxylated flavones) and in the form of heterosides (apigenin, luteolin, eriodictyol) [270]; phenolic acids and derivatives (*p*-coumaric, caffeic, chlorogenic, rosmarinic); tannins; triterpenes; among others, which can serve as alternatives to painkillers medications [269,270,273,274], food products, and cosmetic products, for example, in this last case, with antipruritic and rejuvenating actions [269]. Additionally, these compounds have potential applications in treating various conditions, including skin diseases. The skin is the largest sensory organ in the human body and plays a fundamental role in protecting against continuous environmental aggressions [276]. They can lead to several disorders like chronic pruritus, psoriasis, atopic dermatitis, and other inflammatory conditions [268]. Several studies have shown that formulations containing peppermint exhibited significant improvement in reducing itch intensity due to the ability to inhibit inflammatory pathways by decreasing the release of histamines and other inflammatory mediators [267,268].

It has been used through different routes of administration, topically in myalgias [270] and neuralgias [268,270], itch [270,271], urticaria and eczema, blistering diseases such as herpes zoster, herpes simplex, and chickenpox [271], as an analgesic [267,270], anesthetic, antipruritic [267,268,270], or as a rubefacient [270], but also in perfumery [269]. Via oral administration for liver disorders [270,271], gastrointestinal tract diseases, respiratory infections, irritable bowel syndrome, and as flavoring in food, tobacco, chewing gums/bubble gums, sweets/candies, toothpaste, and mouthwashes. It has been used via inhalation for rhinitis (as a nasopharyngeal decongestant) and sinusitis and flu, with mucolytic and expectorant activities, and due to antiseptic properties [270,271].

Peppermint herbs have been used in traditional medicines for centuries, namely, in Egypt, Greece, and Rome, for gastrointestinal disorders like indigestion, nausea, and common cold and cramps [268]. Growing evidence indicates that peppermint, particularly its menthol component, shows promise as an antipruritic agent. Although more research is needed to optimize its use and while noting limitations in safety for certain groups, *Mentha* × *piperita* could offer a valuable, natural treatment for those suffering from itchy skin conditions [264,267,268,269,271]. In addition, it has shown promise as a source of components for developing new solutions for hair growth, showing the most notable hair growth effects (compared to minoxidil) with induction of a rapid anagen stage [277]. Peppermint has been known for its anti-inflammatory, antimicrobial, and antifungal properties, potent antioxidant activity, and further antiallergenic and antitumor effects [277,278,279]. Even more, there has been arousing interest in the pharmaceutical industry in the development of medicinal and cosmetic formulations in skincare and oral cavity care products due to its wound-healing activities, plus the antimicrobial contribution in anti-acne products and as an eventual alternative in vaginal infections, even as to a protector against dental demineralization and as a caries prevention [278].

#### 4.7.1. Active Compounds

The primary active component in peppermint associated with itch relief is menthol, which comprises about 30–50% of peppermint essential oil. Other compounds such as menthone (10 to 60%), menthyl acetate, cineole, limonene, and others also contribute to its therapeutic effects [264,269,270,271]. Menthol is particularly noted for its cooling and analgesic properties, which can help to soothe skin irritation and reduce itch perception [267,268,278,279]. *Mentha × piperita* extracts and essential oil have notable antioxidant properties, biological actions, and anti-inflammatory activity [278,279]. Specifically, the free radical scavenging effect is due to the presence of monoterpenes such as α-pinene, 1,8-cineole, neo-menthol, borneol, etc., by reducing the cell damage. Researchers also revealed that herbal medicine containing 30–55% menthol and 14–32% menthone has immunomodulatory and antiparasitic properties as observed by the decrease in the plasma level of IL-4 and IL-10, and the reduction in blood eosinophils [278] (Figure 15).

The free flavonoids (strongly methoxylated flavones) and in the form of heterosides (apigenin, luteolin, eriodictyol), the phenolic acids and derivatives (p-coumaric, caffeic, chlorogenic, rosmarinic), and the tannins (6 to 12%) are polyphenols with spasmolytic and carminative actions. Furthermore, its bitter constituents, like triterpenes (ursolic and oleanolic acids), give it a digestive and eupeptic action [269,270]. As demonstrated in Hudz et al. (2023), rosmarinic acid (from the methanolic extract of *M. piperita*) also has a potent antioxidant activity [278]. Another study revealed that phenolic compounds are interesting against the human respiratory syncytial virus, through their antiviral effect [278,279] (Figure 16).

The dominant constituent available in peppermint’s herb is the essential oil, between 0.5 to 4%, whose major component usually is menthol (a form of (−)-menthol), with small quantities of its stereoisomers (+)-neomenthol and (+)-iso-menthol. As a selective marker for peppermint, which is not present in other Mentha species and is a component that exists in smaller quantities (1.0–8.0%), menthofuran stands out. On the other hand, the smaller the amount of menthofuran and pulegone, the better the commercial quality of *Mentha* × *piperita* essential oil [264,278]. According to the concentration of the essential oil used, pro- and antioxidant activities were observed in in vivo tests, although more studies are needed on this item [278].

#### 4.7.2. Mechanism of Action

The metabolism of peppermint essential oil occurs through a reaction with glucuronic acid, and it is subsequently eliminated in urine or feces because it is a set of aromatic constituents with low molecular weights and diverse chemical structures, which can readily cross the cutaneous barrier and reach the bloodstream [279]. Pharmacokinetic studies showed that the main compound is excreted in the urine in the form of menthol β-glucuronide and subsequent investigation confirmed the presence of this metabolite, indicating the existence of two more, the mono- and hydroxylated di-derivatives of the menthol [269]. Overall, the active constituents of the oil can function as penetration enhancers (mainly the menthol) [276,279], promoting transdermal absorption, as they cause a change in the structure of the stratum corneum barrier of the skin and interact with the lipids of the intercellular stratum corneum to improve the diffusivity of osmotic absorption [279].

Menthol’s antipruritic effects are thought to result from its interaction with specific ion channels, primarily the transient receptor potential channels (TRP), such as the transient receptor potential melastatin 8 (TRPM8). Activation of these channels leads to a cooling sensation that counteracts the sensation of itch. Additionally, menthol and other peppermint compounds may inhibit inflammatory responses by reducing the release of histamines and other inflammatory mediators, further helping to relieve pruritus. This modulation of nerve signals and inflammatory pathways suggests a multifaceted approach to peppermint’s action against itchy skin [279].

Therefore, menthol is an agonist of the TRPM8 channel. In turn, this constituent can activate the TRPM8 channel to inhibit the chemical and mechanosensory responses of nociceptive TRP channels and decrease the liberation of pro-inflammatory mediators from nerve endings, and this mechanism of action can occur in the case of irritable bowel syndrome. The menthol in peppermint oil decreases oxidative stress in colon tissue, reducing malondialdehyde levels and the end product of lipid peroxidation [279].

On the other hand, according to Amjadi et al. (2012), Elsaie et al. (2016), and Zhao et al. (2022), the investigators have demonstrated that menthol inhibits pruritus by activating A-delta fibers and k-opioid receptor [267,268,279]. So, as one of the potent compounds of peppermint oil for external use, menthol is a significant constituent in ointment that has cooling and moderate analgesic actions [267]. Once again, as reported by Elsaie et al. (2016), Dr. A. Wright noted the success of using peppermint oil in treating facial neuralgia in the People’s Republic of China and used it in his clinical practice [268].

To reduce inflammation and prevent the development of chronic diseases, it was demonstrated that peppermint extracts and essential oil possess anti-inflammatory properties, whose ethanolic extract of *M. piperita* highly decreased the lipopolysaccharides (LPS)-induced nitric oxide production in a concentration-dependent form and suppressed the production of the proinflammatory cytokines (TNF-α, IL-6, PGE2) in the cells stimulated with LPS compared to the cells stimulated without putting the extract. The phenolic compounds and the various constituents, which act synergistically in the peppermint plant, explain such biological properties. Furthermore, when applying a cream with *M. piperita* essential oil (0.5% *w*/*w*), there was high wound healing in the affected area, better epithelization, fibroblast population, and collagen deposition in the animals treated with it in comparison with the vehicle and Madecassol^®^; these activities are considered to have occurred due to menthol. Also, according to a carrageenan-induced paw edema test, the anti-inflammatory effect of peppermint’s essential oil was demonstrated when administered orally at doses of 20 and 200 mL/kg, thus highlighting the wound-healing and anti-edematogenic impact, again, due to the presence of menthol [278].

Regarding atopic dermatitis and the use of peppermint essential oil, one study in mice with wounds showed the following intracellular mechanisms: inhibition of the expression of molecules in STAT1- and STAT3-dependent pathways, decreasing epidermal thickness and mast cell infiltration, inhibition of lymphocyte proliferation and forkhead box P3 expression (menthol dose-dependently), inhibition action of menthol on Th1 cell differentiation, and subsequent decrease in IFN-γ expression along with T-bet downregulation, with notable wound healing [279].

In another study with transdermal medication of peppermint oil, a significant improvement in psoriasis symptoms was observed in mice. For instance, reduction in pruritus and erythema levels, improvement in skin elasticity and melanin levels, and more body weight gain coexisted with lower serum levels of IL-10 and TGF-β [279].

Several scientific researchers demonstrated that the anticarcinogenic (peppermint extracts) and cytotoxic (peppermint essential oil) properties are due to the influence of different terpenes and/or some active compounds present, among other actions, with induction of apoptosis and interference in the oxidative balance and, due to the lipophilic profile of the essential oil, it can easily cross cell membranes and reach the interior of the cell [278]. On the other hand, formulations with the plants of peppermint (or their respective active compounds) should serve as a starting point for further research and laboratory studies to justify their future use in the treatments of cancer, and even with complementary actions to their medical products [278,279], due to their evidence in the regulation of the expression of some oncogenes, encompassing the epidermal growth factor receptor, promoting the apoptosis, and suppressing the activity of Topoisomerase I to inhibit gene expression in cancer cells, among others [279].

Other studies highlighted that peppermint formulations exhibited hypotensive, vasorelaxant, and antiplatelet properties and, consequently, may reduce the risk of cardiovascular diseases (CVDs, such as high blood pressure, acute myocardial infarction) by the reduction in inflammation, i.e., in the decrease in glycemia, total cholesterol levels, triacylglycerides, low-density lipoproteins indices, urea levels, and blood pressure, weight, among others, and an improvement in high-density lipoprotein cholesterol levels. So, it is a promising source of compounds for future uses in medicinal products with cardioprotective actions, for example, to reduce high blood pressure or to prevent acute myocardial infarction. Such cardioprotective properties may be due to increased expression of plasma antioxidant enzymes and genes involved in Ca^2+^ homeostasis and/or its antiadrenergic actions by rosmarinic acid. In sum, formulations of peppermint possess cardioprotective effects with concomitant relation to its antioxidant effects, enhanced high-density lipoprotein cholesterol indices, and reduction in the arterial smooth muscle tonicity (hypotensive impact with a decrease in the heart rate and systolic blood pressure) [278]. It has demonstrated potent antioxidant activity, effectively scavenging reactive ROS and inhibiting lipid peroxidation [278,279]. Furthermore, *M. piperita* oil, as it has antioxidant properties, decreases melanin synthesis in B16-F10 cells, regulating the expression of the microphthalmia-associated transcription factor (MITF), tyrosinase-related protein (TRP)− 1, TRP-2, and tyrosinase [279].

The role of peppermint compounds in a high spectrum of diseases is well described. It is related to central nervous system diseases [273,275,278,279] (while prevention or treatment possesses neuroprotective action in some neurodegenerative disorders, for instance, in Alzheimer’s and Parkinson’s diseases, which is provided by the inhalation of the peppermint’s essential oil) [278], respiratory diseases [269,270,271,279], cardiovascular protection [271,278], liver disorders [270,271,278,279], gastrointestinal diseases [278,279] (including irritable bowel syndrome and infantile colic pain) [269,270,271,278], antiemetic action [267,270,271,272], antispasmodic effects [278], cancer treatment [278,279], and genitourinary tract diseases [271,278].

Another subject that has been increasing the interest of many authors is skin diseases. Some abundant studies are about chronic pruritus (>6 weeks) due to hepatic, renal, or diabetic causes [268,276,279], Pruritus Gravidarum during pregnancy originated from hormonal variation [267,279], wound-healing [278,279], acne [278], psoriasis, atopic dermatitis [279,281], rejuvenation [269,278], and skin infections [269,270,271,278,279]. Since *M. piperita* induces apoptosis of inflammatory cells, it may expedite the healing process by shortening the inflammatory phase. Additionally, *M. piperita* could enhance collagen synthesis and promote the migration and differentiation of fibroblasts.

Nevertheless, more research and laboratory studies are needed to understand its mechanisms of action and its possible practical use as prevention and/or treatment of various pathologies. This medicinal herb is a promising functional resource for the food, cosmetic, and agricultural fields [278,279].

##### Antipruritic Effect

Pruritus is commonly associated with an irritating sensation that evokes the desire to scratch, severely decreasing the quality of life of affected individuals [137,267,281]. Depending on the cause of itchiness, generally associated with systematic diseases, medication, and external aggressions, the skin may appear typical or exhibit signs of inflammation, roughness, or bumps. Continuous scratching can lead to thickened, raised areas of skin that may bleed or become infected [137]. The leading causes of itchiness are neuropathic, psychogenic, systemic, and dermatologic disorders [268]. Even so, chronic pruritus is the most severe disease associated with pruritus, which influences the day-to-day routine [268,279,281].

In general, itching can be classified as histamine-dependent or histamine-independent. It acts on the nerve C-fibers that perceive pain and itching, and the stimulus is transmitted to the brain as itching and to the nerve endings, causing the release of neuropeptides neurotransmitters [136]. Mast cells function as a “powerhouse”, releasing various allogenic and pruritogenic mediators such as proteoglycans, proteases, leukotrienes, biogenic amines, cytokines, and chemokines. These mediators stimulate reciprocal interactions with specific sensory nerve fibers [181].

It has been shown that IL-31 induces acute itch by acting on IL-31RA in sensory nerves, while IL-4 and IL-13 induce chronic itch by acting on sensory nerves and lowering the response threshold to pruritogenic factors [182].

In sum, pruritus can be transmitted through two receptors: G protein-coupled receptors and downstream TRP channels, and this sensation of pruritus needs TRP signaling, regardless of whether there is histamine mediation or not [281].

In cells, there is a correlation between the transduction and transmission of pain and pruritogenic signals, negatively modulating each other. The entire biochemical process is complex and multifactorial, involving peripheral and central components. At the peripheral level, pain and pruritus are caused by subpopulations of specific nociceptors that recognize and transduce algesic and pruritogenic signals. Although more research is needed, scientific evidence suggests the transient receptor potential vanilloid 1 (TRPV1) thermosensory channel is a center for a large number of pro-algesic and pruritic agents, as TRPV1 appears metabolically coupled to most neural receptors that recognize algesic and pruritic molecules. Therefore, investing in TRPV1 antagonists for cutaneous or local application seems promising for alleviating the problematic symptoms of patients suffering from chronic pain or itching [276].

Precisely, menthol’s antipruritic effects are thought to result from its interaction with specific ion channels, primarily the transient receptor potential (TRP) channels, such as the transient receptor potential melastatin 8 (TRPM8). Activation of these channels leads to a cooling sensation that counteracts the sensation of itch. Additionally, menthol and other peppermint compounds may inhibit inflammatory responses by reducing the release of histamines and other inflammatory mediators, further helping to relieve pruritus. This modulation of nerve signals and inflammatory pathways suggests a multifaceted approach in peppermint’s action against itchy skin. Therefore, menthol is an agonist (activating compound) of the TRPM8 channel [279,281]. In turn, this constituent can activate the TRPM8 channel to inhibit nociceptive TRP channels’ chemical and mechanosensory responses and decrease the liberation of pro-inflammatory mediators from nerve endings [279].

Generally, TRPM8 activity inhibits pruritus rather than inducing it, unlike other TRP channels. Topical cooling has been resorted to decreasing itching, for some itching transduction pathways are delayed, as happens with TRPV1, since cooling reduces nervous excitability and conduction speed. Notwithstanding, cold temperatures and menthol also stimulate sensory neurons that express TRPM8. Additionally, it was observed that cooling effectively suppresses histamine and non-histamine pruritus pathways, which are necessary for activating TRPM8 channels or TRPM8-expressing afferent neurons. Therefore, despite not being pruriceptors, TRPM8 neurons are important in suppressing pruritus, as they are part of a spinal interneuron circuit that includes B5-I neurons. In turn, these inhibitory spinal interneurons receive information from menthol-sensitive afferents and form an itch-suppressing neuropeptide called dynorphin. It was also demonstrated that, in the absence of B5-I neurons, menthol cannot suppress pruritus, revealing the importance of these neurons in the antipruritic action and with the appropriate activation of TRPM8 [281].

The TRP superfamily (e.g., TRPV1 and TRPM8) functions as non-selective cation-permeable channels with Ca^2+^ permeability. In general terms, TRP channels perform as molecular sensors of various physicochemical stimuli, namely, changes in pH and chemical irritants (like pepper, wasabi, mustard, and menthol). They also react to thermal, mechanical, osmotic, and actinic (radiation) signals. Diseases such as dermatitis, psoriasis, notalgia paresthetica, urticaria, prurigo nodularis, herpetic neuropathy, and chemotherapy-induced peripheral neuropathy could be supported by a new medicinal generation of topical thermoTRPs. This innovative strategy could also cover other neural and non-neural receptors. The aim is to modulate the kinetics of esterase hydrolysis, extending from the epidermis to the dermis or deeper sites of action [276]. Only for framing these TRP channels do they participate in the sensory pathways of pruritus, pain, and inflammation because they are sensory molecules with diverse functions and are quite numerous on the skin [282].

On the other hand, some researchers have demonstrated that menthol inhibits pruritus by activating A-delta fibers and k-opioid receptors [267,268,279]. So, as one of the potent compounds of peppermint oil for external use, menthol is a significant constituent in ointment that has cooling and moderate analgesic actions, as it reduces the skin’s temperature [267].

The antipruritic mechanism of peppermint is not entirely clarified; however, it is known that it has antipruritic effects that reduce the release of histamines and other inflammatory mediators [279,281], activates A-delta fibers and k-opioid receptors [267,268,279], and as an immunomodulator, decreases the plasma level of IL-4 and IL-10 and reduces the blood eosinophils (when contains 30–55% menthol and 14–32% menthone) [278]. Also, medicinal plants constitute a promising therapeutic alternative to invest in the modulation of peripheral TRPV1 function [276], which is activated by heating agents and high temperatures [276,282], and in cooling substances that target TRPM8, inclusive already with some topical treatments making progress into clinical trials with anti-itching properties because TRP channels are numerous on the skin [281]. Furthermore, modulation of TRPM8 activity may also be necessary for therapeutic uses: cold treatment is often applied for pain relief, and in some cases, hypersensitivity to cold can lead to cold allodynia in patients suffering from neuropathic pain [282].

#### 4.7.3. Clinical Evidence

Some publications relate to the utilization of peppermint in pruritus, including in some skin disorders. A growing number of clinical studies support peppermint oil’s antipruritic effects, some of which will be presented in Table 4.

**Table 4 plants-13-03515-t004:** Clinical studies regarding *Mentha piperita*.

Type of Study/Pathology	Dosage of Peppermint	Principal Outcomes	Publication
Pruritus in Pregnant Women (*Pruritus Gravidarum*)	Bottle containing 60 mL of peppermint oil 0.5% in sesame oil (topical)	pregnant women who were affected by moderate or severe skin itching participants: were second or third-trimester a significant reduction in itch severity was reported in those using a peppermint-based oil versus a placebo (showing a significant statistical difference: effective decrease in severity of Pruritus Gravidarum) when applied twice daily on the areas of pruritus for 14 days by the case (peppermint oil in sesame oil) and control (placebo with only 60 mL of sesame oil) groups menthol can reduce pruritus caused by histamine by cooling the skin, by activation of A-delta fibers and κ-opioid receptors no observation of any particular side effects on subjects the research was just based on the statements of participants and did not continue with it	[267]
Chronic pruritus (hepatic, renal, and diabetic origin)	5% of peppermint oil (topical)	the patients were instructed to hydrate the skin and then apply the preparations (peppermint or petrolatum) topically by hand; this application was done on the areas of pruritus twice daily for 2 weeks the patients were divided into two groups (a total of 50 patients—26 males and 24 females): group I—25 patients (renal: 8, hepatic: 7, and diabetic: 10), treated with 5% of topical peppermint oil group II—25 patients (renal: 8, hepatic: 8, and diabetic: 9), treated with topical petrolatum as a placebo peppermint oil reduced itch intensity in patients with chronic pruritus compared to a placebo group only four patients who applied the oil in skin folds (groin and axillae) complained of a burning sensation; otherwise, there were no reported side effects menthol in low concentrations demonstrated to be effective without irritant effect the long-term use of menthol for chronic pruritus has not been studied	[268]
Wound-healing properties of the essential oil of *Mentha piperita*	Cream * containing 0.5% (*w*/*w*) *Mentha piperita* essential oil (PEO cream) (topical) * Base cream: sweet almond oil (15–16%), beeswax (3–4%), stearic acid (5–8%), cetyl alcohol (1–1.2%), ceteareth-20 (0.4–0.7%), deionized water (66), glycerine (3.4–5%), trolamine (0.6–0.8%)	each group of rats was anaesthetized with 0.01 mL of Ketamile^®^ (Ketamine chlorhydrate, 50 mg/mL, El Kendi Pharmaceutical, Algiers, Algeria) the back hairs of the animals were removed by shaving were used the circular excision wound model and histological examination. The wounds were created with a surgical blade (a circular wound was formed on the dorsal inter-scapular region of each rat by excising the skin with a 2 cm biopsy punch, and the wounds were left open). For histological examination: at the end of the experiment, a specimen sample of cutaneous tissue was collected from the healed skin of each group of rats a quantity of each test cream was applied topically on the wounded position directly: the rats in the negative group (vehicle) were treated with the cream base only, whereas those in the positive control group were treated with Madecassol^®^ cream the PEO cream, the reference drug (Madecassol^®^ 1% cream, containing Centella asiatica extract as an active ingredient), and the vehicle cream bases were administered topically once a day till the wound was completely healed (day 15) when applying the cream, there was a high wound healing in the affected area, a better epithelization, fibroblast population, and collagen deposition (between the sixth and ninth days of treatment) in the animals treated with it in comparison with the vehicle and Madecassol^®^ creams at the sixth day	[278,283]
Inhibitory effect of peppermint essential oil on psoriasis in mice through transdermal medication	Peppermint essential oil (PEO) in doses of 0.2 mL/cm^2^ and 0.4 mL/cm^2^, with dexamethasone sodium phosphate injection (DSPI) at 50 mg/cm^2^ as a positive control (topical)	first: application of imiquimod cream on depilated skin at 4 cm × 6 cm to cause psoriasis in mice through the transdermal medication of PEO, a significant improvement in psoriasis symptoms was observed in mice. For instance, reduction in pruritus and erythema levels, improvement in skin elasticity and melanin levels, and more body weight gain coexisted with lower serum levels of IL-10 and TGF-β histopathological analysis with H&E staining showed that PEO decreased histopathological features of psoriasis in mice	[279]
Uremic pruritus in hemodialysis patients (chronic kidney disease)	Peppermint essential oil: 1 to 2 drops in the area of itching (topical)	the total sample was 98 participants (patients with chronic kidney disease in Haji Adam Malik General Hospital Medan) inclusion criteria: hemodialysis therapy twice a week, no allergies to the aromatherapy given, mild to severe pruritus, without other dermatological problems than pruritus, aged 18–65 years, and no opened wounds in the intervention application area the research was conducted from April to May 2019 the results demonstrated that the uremic pruritus scale reduced to a mild degree of 51.0% through the Wilcoxon and Mann–Whitney statistical tests, it was showed that the intervention group had a *p* value of 0.000 and in the control group there was a *p* value of 0.102, therefore was verified a decrease in uremic itching by the action of peppermint oil (especially due to the menthol, which can provide a cold sensation on the skin that can act as an antihistamine) the duration experienced by the majority took place in less than 6 h, but with a positive impact on uremic pruritus in respondents experiencing chronic kidney disease undergoing hemodialysis therapy	[284]
Synthetic TRPM8 agonist (Cryosim-1) gel for itch (eczema, chronic urticaria, and postherpetic neuralgia with pruritus)	Gel Cryosim-1 (1-diisopropylphosphorylheptane, Chemical Abstracts Service registry no. 1487170-15-9; Dong Wha Pharmaceutical, Seoul, South Korea): cooling mimetic agent compared to vehicle-only gel on itch (topical)	the effects of physical cooling and of cooling mimetic agents (menthol) on itch are attributed to the activation of an ion channel designated as transient receptor potential melastatin 8 (TRPM8) the limitations of menthol, its irritancy and short duration of action, led to the creation of Cryosim-1 Cryosim-1 activates hTRPM8 at a half maximal effective concentration of approximately 0.7 μmol/L and is inactive on TRPA1 and TRPV1 a topical cooling gel demonstrates that control of pruritus can be fast the drug effect occurs within 10 min after application, and an antipruritic action persists at 2 h after administration significant decrease in pruritus two hours after Cryosim-1 application, particularly in patients with urticaria Cryosim-1 is more effective for improving quality of life, giving rapid relief of itch and making it a valuable addition to the treatment of pruritus	[281,285]

While these studies indicate peppermint’s potential, further large-scale, long-term studies are necessary to confirm its effectiveness and clarify optimal application methods.

Additionally, more research should be accomplished to evaluate topical TRPM8 agonists in treating certain types of itching, as it could worsen pruritus in some disorders such as psoriasis [281]. Interesting data, in the temperature-controlled perfusion system subject to cold temperatures ranging from 23 °C to 10 °C, were found showing cells that overexpress TRPM8 have higher levels of intracellular calcium. Activation and modulation occur by menthol reinforcing that TRPM8 indeed functions as a cold-sensitive channel in vivo, as demonstrated when analyzing the behavior of a subset of isolated DRG neurons (the sensitivity to menthol depends on temperature) [282].

#### 4.7.4. Usage and Dosage

According to the German Commission E Monograph, peppermint leaves can be used in infusions for gastrointestinal disorders, irritable bowel, and respiratory affections, with a proposed daily dose of 3 to 6 g [269]. Regarding its essential oil, a daily dose of 0.6 mL in an enteric-coated capsule in the irritable colon is recommended to prevent heartburn [269,270].

Different forms of administration and respective dosages are possible, considering an average daily dose of 6 to 9 g of drugs or preparation with equivalent content (Table 5 and Table 6) [270].

Topical applications of peppermint essential oil diluted to about 1–5% concentration are common for treating and managing itchy skin. This concentration provides adequate menthol for antipruritic action while minimizing potential skin irritation. Application frequency varies, but studies commonly recommend applying a thin layer to the affected area 1–2x/d. For use in pregnancy-related pruritus, formulations designed specifically for sensitive skin are recommended (typically, 0.5%) [267,268,278,279,283,284].

There are some limits to the use of peppermint essential oil in patients with special conditions, including pregnant women—up to 10% via olfactory use; breastfeeding women—up to 10% via olfactory use; neurologically affected patients—up to 20% via the cutaneous use, up to 10% via the oral or sublingual uses, and up to 10% via olfactory use [271].

It is generally advised to dilute peppermint essential oil with a carrier oil, like sesame oil, or a base cream, before applying it to the skin to prevent irritation. These diluted preparations are applied once or twice daily and are effective in short-term itch relief.

External use of peppermint hydrolate (INCI: *Mentha piperita* Leaf Water) is more common due to its gentle and versatile properties. It can be applied directly as a spray mist on affected areas or incorporated into a soothing cream formulation for pruritus or itchy skin. It may also be combined with 0.1–0.5% p/p peppermint essential oil to enhance menthol effects, with recommended application once or twice daily for short-term relief. For sensitive skin or during pregnancy, the diluted hydrolate without essential oil offers a safe and refreshing solution for itching or mild inflammation. To support respiratory function or relaxation, the hydrolate can be misted into the air or onto linens for aromatherapeutic benefits. In hair care, it serves as an effective rinse or scalp mist to reduce itchiness and soothe irritation [286].

#### 4.7.5. Safety and Side Effects

There are reports that peppermint essential oil should not be administered in the case of obstructed bile ducts, gallbladder inflammation, and hepatitis [269].

It is common in French literature, particularly by aromatherapist Dominique Baudoux [271], that peppermint essential oil is contraindicated during pregnancy, which is not portrayed according to other renowned aromatherapists, such as Robert Tisserand and Hervé Staub [264,283]. This restriction, defended by author Dominique Baudoux, is because ketones are present in more significant quantities in the essential oil, such as menthone, isomenthone, pulegone, and piperitone, with no consensus among the various authors regarding their use in pregnant women. Also, its use in babies and breastfeeding women is not recommended, as menthol (terpenic alcohol) and menthone (terpenic ketone) present risks for babies under 30 months years old and for the fetus. During lactation, there are not enough data to ensure their respective safety in this sensitive group, and no toxic effects have been announced from the use of peppermint during pregnancy and lactation [267,283]. Furthermore, it is not recommended for hypertensive patients on protocols lasting more than 15 days. Therefore, the toxicological data point (LD50) for the oral pathway is 4.40, and for the dermal pathway is 5.00 [271].

According to Cunha et al. (2012), it is not recommended that essential oil be used internally during pregnancy, breastfeeding, children under 6 years old, patients with hypersecretory dyspepsia, severe intestinal diseases, and neurological diseases [270].

Nervousness, insomnia, and contact dermatitis may occur in more sensitive people. When inhaling the essential oil, spasms of the larynx and bronchi may appear, especially in children [270]. The menthol can cause incoordination, confusion, and delirium when 5 mL (35.5% peppermint oil) is inhaled during a time-long period [264]. Also, it is necessary to carry out a tolerance test before inhaling, inhaling for 15 s and waiting 30 min [270]. Other probable side effects are allergic reactions, heartburn, nausea, vomiting [283], ataxia, and convulsions (dose-dependent in rats) [264].

The safety summary for peppermint essential oil is hazards—choleretic, neurotoxicity, mucous membrane irritation (low-risk); contraindications (all routes)—cardiac fibrillation, glucose-6-phosphate dehydrogenase (G6PD) enzyme deficiency (especially in people of Chinese, West African, Mediterranean, or Middle Eastern origin, who usually have abnormal blood reactions to at least one of these medicines, or will have been warned to avoid them, such as antimalarials, sulfonamides, chloramphenicol, streptomycin, aspirin), do not use internally or to or near the face of infants or children because of its possible bronchospasm effect; oral contraindications—cholestasis; oral cautions—gastroesophageal reflux disease and hiatal hernia; maximum adult daily oral dose—152 mg; maximum dermal use level—5.4%. This author, at the oral and dermal level, established the following limits and restrictions: 8.0% menthofuran and 3.0% pulegone content, with limits of 0.2 mg/kg/day and 0.5% for menthofuran, and of 0.5 mg/kg/day and 1.2% for pulegone [264,283].

As the regulatory guidelines state, peppermint essential oil has been classified as “Generally Recognized as Safe” (GRAS). The Commission E Monograph authorizes 5–20% in oily and semisolid formulations, 5–10% in aqueous-alcoholic formulations, 1–5% in nasal ointments, and, as an average daily oral dose, 6–12 drops (or 0.6 mL in enterically coated capsules). Since the pulegone level in the essential oil does not exceed 1.0%, peppermint oil is considered safe to use in cosmetic preparations, according to the Cosmetic Ingredient Review Expert Panel. The CEFS of the Council of Europe has established a group TDI for menthofuran and pulegone of 0.1 mg/kg bw [264]. According to the US Food and Drug Administration, peppermint essential oil is generally assumed to be safe. However, higher doses used for medicinal purposes have not been widely regulated, as this essential oil is sold as a dietary supplement [268].

In analyzing adverse skin reactions, no adverse reactions were recorded by using 1% peppermint oil in a 48 h occlusive patch test on 380 eczema patients. On the other hand, for pharmaceutical products with peppermint oil included, urticarial hypersensitivity has occurred, but these reactions are uncommon and normally become skin sensitivity historically. In the case of tests using 2% on successive dermatitis patients, peppermint oil caused reactions in 3 (0.25%) of 1200, 1 (0.5%) of 200, 9 (0.6%) of 1606, and 2 (0.6%) of 318. Moreover, 2% peppermint oil produced a reaction in 1 (0.13%) of 747 dermatitis patients suspicious of fragrance allergy, and Germany’s IVDK research revealed 42 (0.64%) of 6546 dermatitis patients with the same conditions. However, another Finland study did not register irritant or allergic reactions to peppermint oil in 73 dermatitis patients. Other reports were recorded: 1781 with contact dermatitis of 18,747 dermatitis patients, and 1 (0.005% of dermatitis patients and 0.06% of contact dermatitis patients) was allergic to peppermint oil. Also, 75 (1.4%) were allergic to cosmetic products of the 1781 patients. In a 1985 study, 12 workers in a food factory developed hand eczema, and just 1 reacted to 2% peppermint oil. An interesting fact, a 65-year-old aromatherapist with diverse essential oil sensitivities was reported to react slightly to 5%, but without any reaction to 1% peppermint oil. In another case, five years ago, a man spilled acid on the back of one hand and needed skin grafts, and when handling undiluted peppermint oil at work, he spilled some on the same hand, observing that the peppermint oil created substantial necrosis, and the fresh skin grafting was successful. Peppermint oil proved not to be phototoxic, according to an in vitro test [264].

Additional organ-specific effects: oral mucous membrane sensitivity due to peppermint oil and menthol contact or after excessive extended utilization, such a burning feeling, ulceration, and inflammation, but with very rare reactions; cardiovascular effects, among other reports, mainly due to menthol cigarettes addiction with possible cardiac fibrillation (interaction with the quinidine medicine for heart rhythm regulation) and bradycardia, although there is no direct evidence that the oil poses any risk to health, and very slight inhibition of platelet aggregation originating from menthol and menthone; neonatal toxicity that results from menthol’s accumulation due to the deficiency of the G6PD enzyme causing neonatal icterus, because this enzyme generally is involved in detoxification of menthol; neurotoxicity with some dose-related lesions in rat cerebellum at 40 and 100 mg/kg/day, but without any effect at 10 mg/kg/day, proving that high doses of this essential oil can cause neurotoxicity; hepatotoxicity with the observation of liver toxicity in rats due to the use of massive doses of menthol or menthone (above 200 mg/kg po for 28 days), liver and lung toxicity in mice through menthofuran, and in rats, oral doses at 250 mg/kg/day for 3 days leading to hepatotoxicity, also β-pulegone is hepatotoxic; among other organ-specific effects [264].

In general, due to the low molecular weight of essential oils, they can easily cross the cutaneous barrier, which will be absorbed into the skin within 20 to 50 min, depending on their chemical and physical nature. In this context, using high concentrations of peppermint essential oil can produce toxic and even lethal responses, being associated with interstitial nephritis and acute renal failure [279,283]. Remarkably, the oral administration of peppermint oil is recommended with caution in human beings: internal use in rats at the dosages of 40 and 100 mg/kg body weight per day for 28 days may cause histopathological change, including cyst-like spaces scattered in the white matter of the cerebellum, but without encephalopathy; subchronic toxicity researching in rats with internal use at the dosage of 100 mg/kg body weight per day for 90 days causes cyst-like spaces scattered in the white matter of the cerebellum in male and female rats and no encephalopathy, but with nephropathy in male rats; internal use in rats at the dosages of 400 and 800 mg/kg b.w./day for 28 days can decrease creatinine, increase alkaline phosphatase activity and bilirubin in plasma, raise the weight indices of liver and spleen, and cyst-like spaces in the white matter of the cerebellum were demonstrated histopathologically. However, in a short-term toxicity research of menthone in rats with a low dosage of 200 mg/kg b.w./day, it was not verified that there was any adverse effect [279].

Further, some drug interactions with peppermint oil [264]: felodipine orally administered for hypertension involving cytochrome P3A4 (CYP3A4) enzyme activity can increase its bioavailability; even though it is not completely enlightened, it was found in a small research study on rats that a dose of 100 mg/kg of essential oil tripled the bioavailability of an immune-suppressant drug such as cyclosporin; other variable effects were observed with pentobarbitone, codeine, and midazolam from the oral intake at 0.1 or 0.2 mL/kg. However, the results strongly demonstrated that a human therapeutic dose of this essential oil (even at up to 1.2 mL/day) would not be sufficient to interact with these medicines; also, the oil can raise the permeability of rat skin to 5-fluorouracil, but in human skin, it reduced the permeability of benzoic acid.

Peppermint essential oil represents a low-risk skin allergen and does not require restrictions on its use in reactive skin. On the other hand, it can be neurotoxic in excessive doses, which seems to be influenced by the content of menthone and pulegone in oil, and extra attention must be paid to nasal instillation in young children to avoid referred neurotoxicity. Nowadays, the pulegone contents for commercial peppermint oils are safely traded, at least in the Occidental world, with some of them containing less than 1% pulegone (depending on the type of soil, the time of picking, and other factors) [250].

The plant’s essential oil is commonly studied for its potential to alleviate itch (pruritus) associated with skin conditions, thanks to its cooling and numbing effects, which can decrease the urge to scratch and relieve itchiness. The key active compounds in peppermint that are responsible for their therapeutic effects include menthol, menthone, and various flavonoids. Menthol, the primary constituent, is particularly noted for its cooling effect, which is thought to contribute significantly to its anti-pruritic actions by activating cold-sensitive receptors in the skin (TRPM8 receptors), and it has been shown to provide temporary itch relief without causing local inflammation.

For those whose topical and systemic treatments tend to be irritating, contraindicated, or less tolerated, the cutaneous treatment for itching with peppermint oil is more acceptable, effective, and with great results (in recommended concentrations), easy-to-use, safe, low-priced, and has a pleasant odor [268]. From this perspective, this medicinal plant constitutes a promising investment for discovering, developing, and improving new and innovative medicines [283]. Although Mentha and peppermint oil have several therapeutic properties and few side effects, the possible dose-dependent toxicity resulting from the essential oil is a public health concern. Concerning essential oil and its safe use, for example, in the most diverse areas of health and food, it is also important to better elucidate the toxic dosage under a specific condition and its toxic mechanism [279]. Due to potential toxicity if ingested in large amounts, topical use probably is the preferred method for pruritus treatment.

Additionally, menthol should be used cautiously in children and those with known allergies to mint. Pregnant and breastfeeding women should consult with healthcare providers before use due to limited data on high-dose safety. Further research is justified in exploring peppermint herb potential as a long-term agent.

### 4.8. Oenothera biennis

*Oenothera biennis* (Figure 17), known as evening primrose, is a biennial plant native to North America and Europe [287]. It belongs to the Onagraceae family and is widely recognized for its medicinal properties, particularly its seeds, a rich source of essential fatty acids like gamma-linolenic acid (GLA) [288]. Traditionally, evening primrose has been used to address various skin conditions, particularly inflammation and itching, such as eczema and atopic dermatitis [288,289]. Interest in *O. biennis* stems from its potential anti-inflammatory and antipruritic effects, making it a promising complementary treatment for pruritus associated with skin conditions [290].

#### 4.8.1. Active Compounds

The seeds of *O. biennis* are recognized for their rich composition of fatty acids, with GLA being the primary active component [290]. GLA, which accounts for around 7–14% of seed oil, is converted in the body to dihomo-γ-linolenic acid (DGLA) [292]. DGLA subsequently produces prostaglandin E1 (PGE1), a compound with anti-inflammatory properties that help soothe skin inflammation [293,294]. Another significant seed oil component is linoleic acid, an essential fatty acid crucial for preserving skin barrier function and reducing water loss through the skin [290,295,296]. In addition to these fatty acids, *O. biennis* contains phenolic compounds and flavonoids, which offer antioxidant benefits by mitigating oxidative stress within the skin [297,298,299,300]. Together, these compounds contribute to the plant’s effectiveness in addressing pruritus associated with inflammatory skin conditions (Figure 18).

#### 4.8.2. Mechanism of Action

The therapeutic effects of *Oenothera biennis*, particularly its seed oil, are primarily attributed to its GLA content [296]. When ingested or applied topically, GLA is metabolized into DGLA, a precursor to anti-inflammatory eicosanoids, such as PGE1. These compounds reduce inflammation and calm the skin by inhibiting inflammatory pathways, such as leukotrienes and interleukin production, which are often overactive in skin disorders associated with itching. By enhancing skin barrier integrity, *O. biennis* oil also helps maintain hydration, essential in reducing transepidermal water loss—a common issue in pruritic and inflamed skin conditions like eczema [302,303]. Additionally, the antioxidant effects of the plant’s phenolic compounds combat oxidative stress, further supporting skin health and reducing irritation.

##### Antipruritic Effect

GLA supports skin barrier function by replenishing essential fatty acids in the skin’s lipid matrix, reducing transepidermal water loss, and enhancing moisture retention, which helps alleviate dryness-induced itch. The antioxidant properties of the phenolic compounds and flavonoids in evening primrose oil (EPO) also play a role in reducing oxidative stress in inflamed skin, which can exacerbate itching. Additionally, by stabilizing cell membranes and decreasing the release of pruritogenic (itch-inducing) mediators, EPO further helps to soothe irritated skin. Clinical studies on EPO have shown reductions in pruritus severity in conditions like eczema, with some patients experiencing relief after several weeks of consistent use, although individual responses can vary.

#### 4.8.3. Clinical Evidence

Research on the effects of *Oenothera biennis* in pruritus management has produced mixed results, particularly in clinical trials with patients suffering from atopic dermatitis and other inflammatory skin conditions [290,304]. Some studies have demonstrated that supplementation with EPO can reduce symptoms of atopic dermatitis, including pruritus, dryness, and overall disease severity [305,306,307]. However, other studies report minimal to no significant improvement, suggesting individual variability in response to EPO [307]. For instance, in controlled studies where patients with eczema were administered EPO orally over several weeks, some observed a marked reduction in itching intensity, while others did not experience noticeable relief [308,309]. The inconsistent results indicate that while EPO may benefit specific populations, further research with larger sample sizes and standardized dosages is required to establish its effectiveness and identify the subgroups most likely to respond positively to the treatment.

#### 4.8.4. Usage and Dosage

EPO from *Oenothera biennis* is commonly available in oral and topical formulations [310]. For oral use, doses generally range from 1000 to 3000 mg daily, divided into two or three doses, depending on individual needs and response [290,310,311]. This dosage has been typically used in clinical settings to assess its effects on pruritus and skin health. Topically, EPO can be applied as creams, oils, or lotions to affected areas, usually twice daily [290,312]. Topical application is considered safe for localized pruritus, as it delivers GLA directly to the skin. It is generally recommended to consult a healthcare professional before starting supplementation, especially for individuals with underlying conditions or those on other medications. Results may take several weeks to become evident, as fatty acids in EPO accumulate gradually in cell membranes and exert their anti-inflammatory effects over time [290]. Principal constituents and methods of use of evening primrose are presented in Table 7.

**Table 7 plants-13-03515-t007:** Principal Constituents and Methods of Use.

Constituent	Part of Plant	Method of Extraction	Common Uses	Reference
Gamma-Linolenic Acid	Seeds	Cold-Pressed Oil	Oral supplements, topical creams	[313]
Linoleic Acid	Seeds	Cold-Pressed Oil	Skin health, barrier function	[303]
Phenolic Compounds	Leaves, Stems	Maceration, Infusion	Antioxidant formulations	[314]
Flavonoids (e.g., quercetin)	Leaves, Flowers	Maceration, Infusion	Anti-inflammatory applications	[201]

Cold-Pressed Oil—Extracted from seeds; retains fatty acids. Maceration—Soaking plant parts in a solvent to extract compounds. Infusion—Steeping plant materials in hot water for topical use.

#### 4.8.5. Safety and Side Effects

EPO from *Oenothera biennis* is generally considered safe, though some individuals may have side effects [290]. Common adverse effects include mild gastrointestinal issues such as nausea, diarrhea, and stomach discomfort [315,316]. In rare cases, headaches have been reported [315]. Allergic reactions are infrequent but may include skin rash, itching, or localized swelling, especially with topical application [317]. There is a potential risk for individuals with seizure disorders, as EPO may lower the seizure threshold in sensitive individuals [318]. Interaction with anticoagulant medications is also possible, as GLA may increase bleeding risk when combined with blood thinners [319]. Pregnant or lactating women, as well as those with preexisting health conditions, should consult a healthcare provider before use, as data on safety in these populations are limited [320]. Overall, EPO has shown good tolerability, particularly in lower doses, but careful monitoring is advisable for prolonged or high-dose use.

## 5. Discussion

Pruritus is a complex symptom that significantly impacts the quality of life, particularly for patients in palliative care. This review highlights the therapeutic potential of eight medicinal plants—chamomile, aloe vera, calendula, curcumin, lavender, licorice, peppermint, and evening primrose—in managing pruritus. These findings are particularly relevant in palliative care, where symptom management is central to improving patient comfort and well-being.

The selected plants exhibit diverse mechanisms of action that align with the multifactorial nature of pruritus. For instance, chamomile and calendula provide anti-inflammatory and skin-soothing effects, while aloe vera enhances hydration and promotes barrier repair. Lavender and peppermint deliver immediate relief through cooling and analgesic properties mediated by bioactive compounds such as menthol and linalool. With its potent anti-inflammatory and antioxidant properties, Curcumin shows promise in addressing cytokine-mediated pruritus, a common issue in systemic conditions like cancer or chronic kidney disease. Evening primrose oil, rich in gamma-linolenic acid, supports the management of atopic dermatitis-associated itching, while licorice reduces inflammation and histamine release, targeting allergic pruritus.

In palliative care, where conventional treatments such as antihistamines, corticosteroids, or immunomodulators may be limited by side effects or diminishing efficacy, these medicinal plants offer a complementary approach. Their broad safety profiles and natural origins resonate with the holistic philosophy of palliative care, addressing not only physical discomfort but also the psychological and emotional distress associated with chronic pruritus.

Antigenotoxic analysis of several medicinal plants discussed in this review has shown promising results, particularly in palliative care. These studies have demonstrated the potential of various plant-derived compounds to counteract genetic damage induced by oxidative stress, a common concern in palliative patients undergoing chemotherapy or other treatments that may lead to DNA damage [321,322]. The antigenotoxic properties of plants like curcumin, aloe vera, and licorice suggest they could play a beneficial role in mitigating cellular damage, improving overall patient health, and enhancing quality of life [323,324,325]. While more extensive studies are needed, these findings highlight the therapeutic potential of plant-based therapies as adjuncts in palliative care, offering hope for reducing the long-term genetic consequences of cancer treatments and other conditions associated with palliative care.

However, the review also underscores critical gaps in the current research. Standardized dosages, formulations, and long-term safety profiles remain poorly defined for most plants. Additionally, robust clinical trials are needed to validate preclinical findings and determine their efficacy in palliative care populations. Incorporating these plants into evidence-based care protocols requires collaboration between traditional botanical knowledge and contemporary scientific research.

Integrating medicinal plants into palliative care can enhance symptom management, reduce reliance on pharmacological interventions, and align treatment approaches with patient preferences for natural therapies [326]. This direction aligns with the growing emphasis on personalized, patient-centered care, particularly for individuals with life-limiting conditions [327]. Medicinal plants can play a transformative role in treating pruritus in palliative care settings by bridging ancient botanical practices with modern medical evidence [328,329].

### 5.1. Potential Interactions Between Medicinal Plants and Standard Palliative Care Drugs

When incorporating medicinal plants into palliative care, an important consideration is the possibility of interactions with standard pharmaceuticals. These interactions can affect the efficacy or safety of treatments. Some plant-based compounds may interfere with the metabolism of conventional drugs, altering their concentration and potential side effects. For example, certain plants such as licorice (*Glycyrrhiza glabra*) may affect corticosteroid metabolism, and peppermint (*Mentha piperita*) may influence the absorption of medications through its effects on the gastrointestinal tract [330,331]. It is essential that patients using plant-based therapies in conjunction with conventional drugs be monitored for potential interactions, and healthcare providers should be consulted before integrating these therapies into treatment plans.

### 5.2. Challenges in Incorporating Plant-Based Therapies into Daily Medical Practice

Incorporating plant-based therapies into daily palliative care practices presents several challenges. One of the most significant hurdles is resource availability. Not all healthcare settings may have access to high-quality medicinal plants or the necessary formulations, such as essential oils, extracts, or hydrolates, which can limit their widespread use [9,332]. Furthermore, costs can be significant, especially in resource-limited settings where more affordable pharmaceutical options may be prioritized.

Training healthcare professionals in the use of plant-based therapies is another critical challenge [333]. Proper knowledge of safe usage, potential side effects, and interactions with standard drugs is necessary to ensure these therapies’ effective and safe use. To facilitate the integration of these treatments, further clinical research and standardized guidelines are needed to address the practicalities of including plant-based therapies in palliative care.

### 5.3. Ethical and Cultural Considerations in Plant-Based Medicine

Ethical and cultural aspects play an essential role in accepting and using plant-based therapies, particularly in diverse patient populations. Different cultures may have varying familiarity with and acceptance of herbal medicine [334]. It is essential to approach these therapies with cultural sensitivity, ensuring that patients’ beliefs and preferences are respected. Additionally, informed consent is a key ethical consideration [335]. Patients must be fully aware of the potential benefits, risks, and alternatives to plant-based treatments. In palliative care, where the goal is to improve quality of life, it is crucial to tailor therapies to each individual’s cultural background and values, ensuring that they feel comfortable and empowered in making decisions about their treatment.

Incorporating plant-based therapies into palliative care also raises questions about equity and access [336]. Not all patients may have equal access to these therapies due to financial, geographical, or healthcare infrastructure limitations. Addressing these disparities is crucial to ensure all patients can equitably access the potential benefits of plant-based medicine.

### 5.4. Limitations

While this review provides valuable insights into the potential of medicinal plants for pruritus relief in palliative care, several limitations must be acknowledged. One notable weakness is the lack of standardization in the methodologies across the studies reviewed. The variation in study designs, dosages, and formulations of plant-based therapies makes it challenging to draw definitive conclusions about the most effective treatments. Moreover, the diversity of data available, ranging from preclinical studies to clinical trials, adds complexity to the assessment of the efficacy of these therapies.

Additionally, while the review highlights the promising potential of medicinal plants, there is insufficient discussion of the limitations of the current knowledge base. Many studies lack rigorous control groups, and some are limited by small sample sizes, making it difficult to generalize results across larger populations. The heterogeneity in plant species, preparation methods, and outcome measures further complicates the comparison of results.

The mechanisms of action of the plants discussed in this review are often inferred from preliminary studies, but further experimental data are required to understand better how specific bioactive compounds interact with pruritus pathways. The current evidence on the mechanisms of action is primarily based on in vitro and animal studies, which may not always reflect the complexity of human physiology.

### 5.5. Future Research

While the therapeutic potential of medicinal plants for managing pruritus is promising, further research is necessary to bridge the gap between traditional knowledge and clinical application, particularly in palliative care settings. Key areas for future investigation include standardization, clinical trials, mechanisms of action, and integration into care protocols.

One priority is the standardization of formulations and dosages. Current studies often use varying preparations of medicinal plants, including raw extracts, essential oils, and topical applications, making it difficult to compare findings or recommend specific treatment protocols. Establishing standardized concentrations, extraction methods, and application guidelines will enhance reproducibility and clinical utility.

Rigorous clinical trials are essential to validate the efficacy and safety of these plants, especially in palliative care populations with complex medical conditions and unique vulnerabilities. Randomized controlled trials with diverse patient cohorts are needed to assess outcomes such as itch intensity, skin barrier function, and overall quality of life. Long-term studies will also help identify potential adverse effects and establish chronic-use safety profiles.

Exploring the mechanisms of action of these plants at molecular and cellular levels is another critical avenue. While many plants’ anti-inflammatory, antioxidant, and antipruritic properties are recognized, their specific interactions with pruritus pathways—such as cytokine modulation, sensory nerve desensitization, and skin barrier restoration—require further elucidation. This research will provide insights into tailoring plant-based therapies to specific types of pruritus.

Finally, integrating medicinal plants into palliative care protocols demands interdisciplinary collaboration. Future research should address how these therapies can complement existing treatments, considering their accessibility, cost-effectiveness, and cultural acceptability. Investigating patient and caregiver perspectives on using natural remedies will also inform guidelines that align with patient-centered care principles.

By addressing these research gaps, the field can advance toward evidence-based integration of medicinal plants, offering safe, effective, and holistic approaches for managing pruritus in palliative care.

## 6. Conclusions

Pruritus remains a significant challenge in palliative care, where symptom management is essential for improving quality of life. This review underscores the potential of medicinal plants such as chamomile, aloe vera, calendula, curcumin, lavender, licorice, peppermint, and evening primrose as complementary treatments for pruritus. These plants exhibit diverse pharmacological properties, including anti-inflammatory, antioxidant, and skin-soothing effects, aligning with pruritus’s multifactorial nature. Their broad safety profiles and natural origins make them particularly appealing in holistic and patient-centered care approaches.

Integrating medicinal plants into palliative care requires a multidisciplinary effort that combines traditional botanical knowledge with modern scientific evidence. By addressing research gaps and developing evidence-based guidelines, medicinal plants could serve as a practical, accessible, and culturally sensitive option for managing pruritus and enhancing the overall comfort of patients in palliative care. This approach offers relief from a distressing symptom and aligns with the principles of holistic care, addressing both physical and emotional well-being.

## Figures and Tables

**Figure 1 plants-13-03515-f001:**
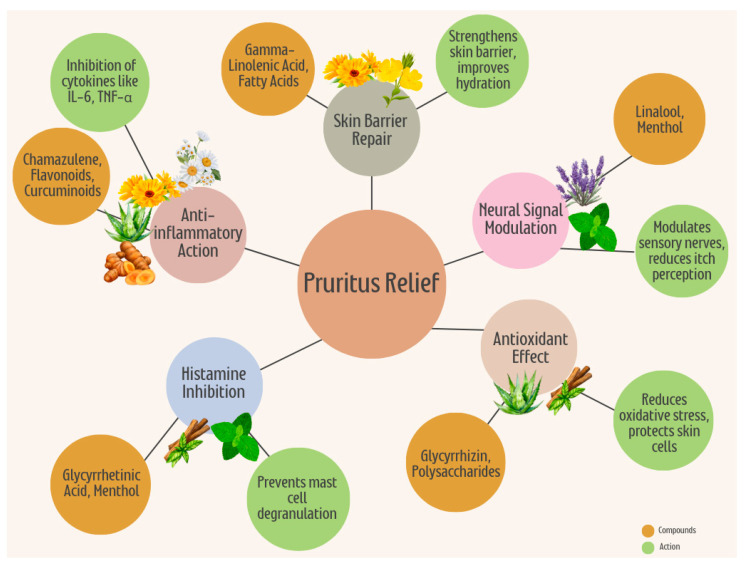
Mechanism of action and active compounds of the medicinal plants.

**Figure 2 plants-13-03515-f002:**
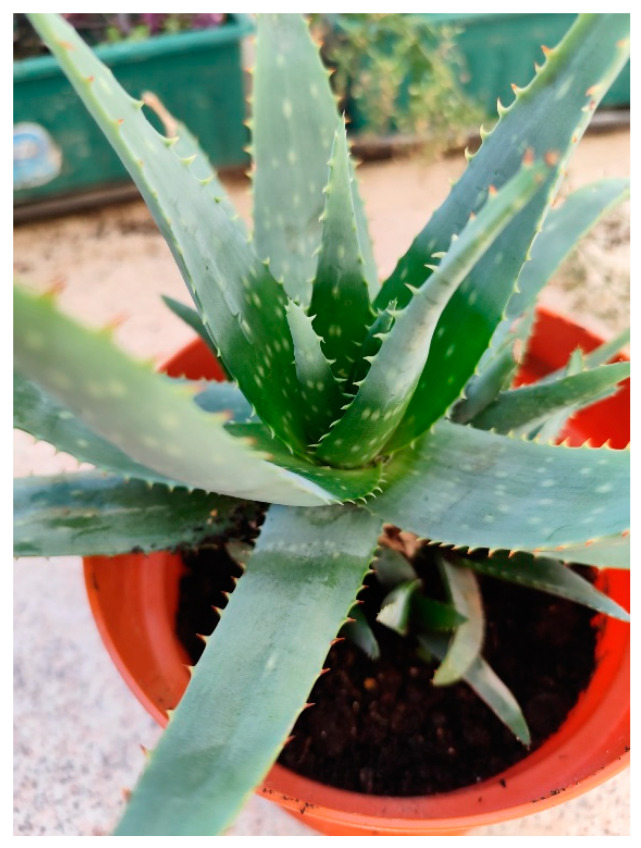
*Aloe barbadensis*.

**Figure 3 plants-13-03515-f003:**
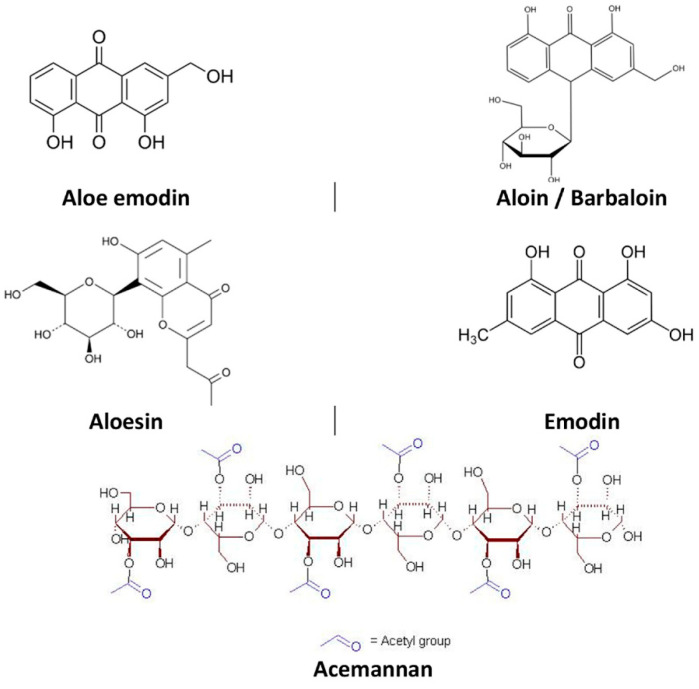
Chemical structure of compounds isolated from *Aloe barbadensis* with pharmacological activity. Adapted from Sánchez et al. (2020) [37].

**Figure 6 plants-13-03515-f006:**
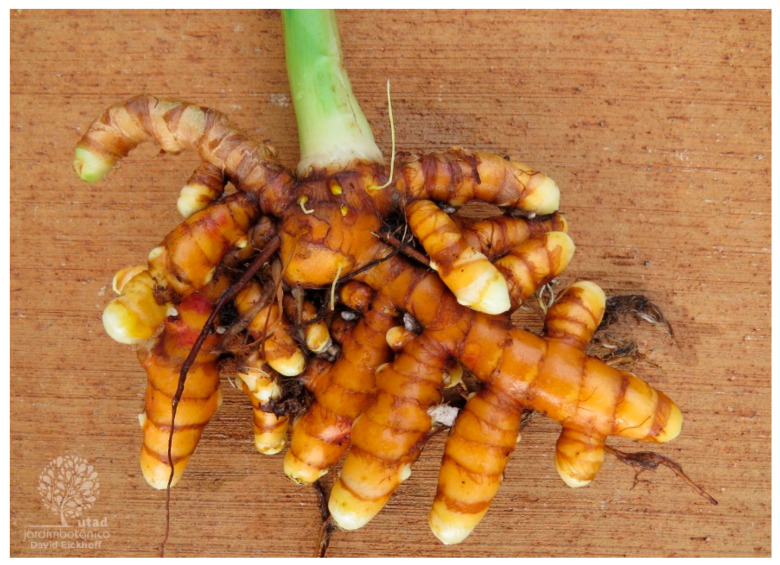
*Curcuma longa*. Adapted from Jardim Botânico da UTAD [128].

**Figure 7 plants-13-03515-f007:**
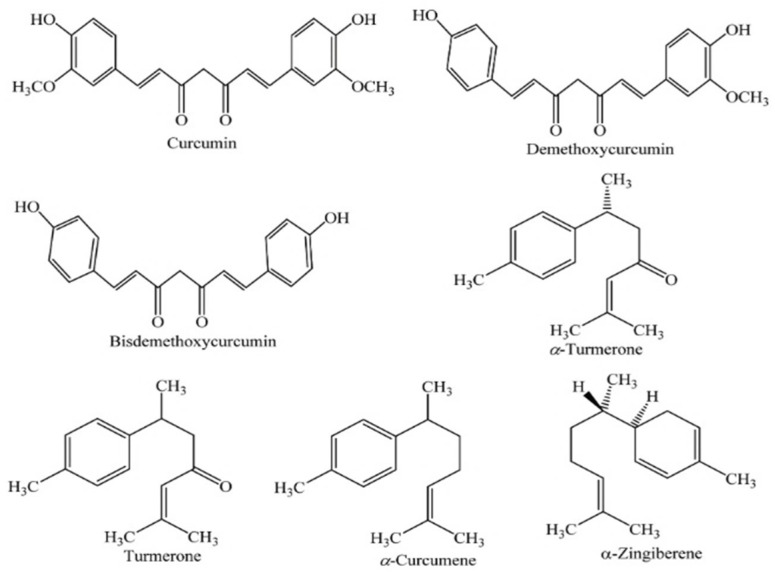
Major chemical active compounds in turmeric. Adapted from Shahidi, 2018 [150].

**Figure 8 plants-13-03515-f008:**
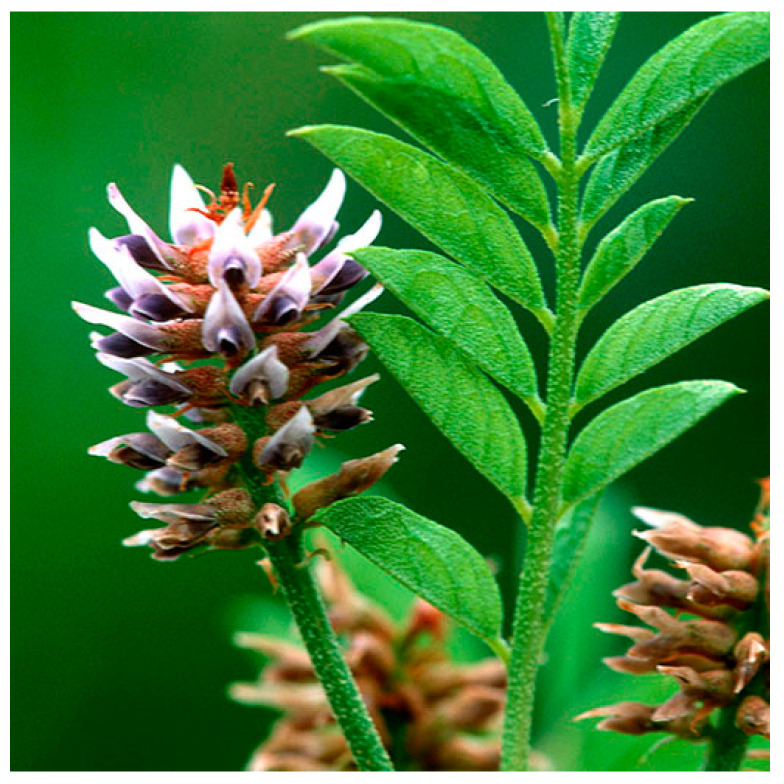
*Glycyrrhiza glabra*. Adapted from the National Center for Complementary and Integrative Health [197].

**Figure 9 plants-13-03515-f009:**
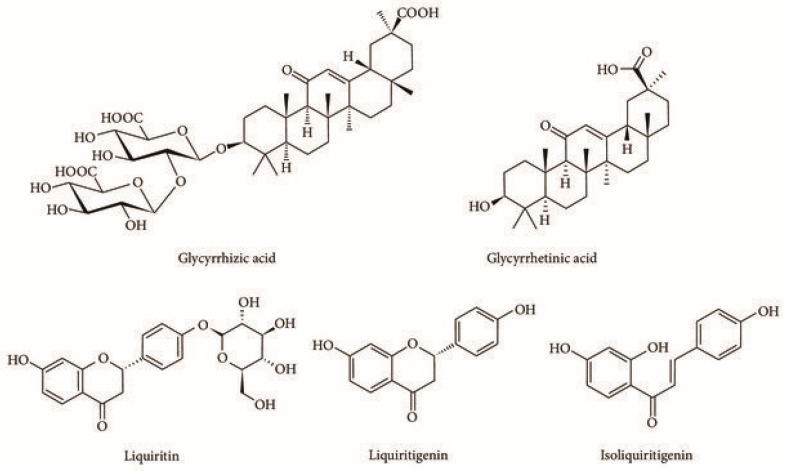
Five representative active compounds in licorice (glycyrrhizic acid, glycyrrhetinic acid, liquiritin, liquiritigenin, and isoliquiritigenin). Adapted from Cao et al. (2017) [204].

**Figure 10 plants-13-03515-f010:**
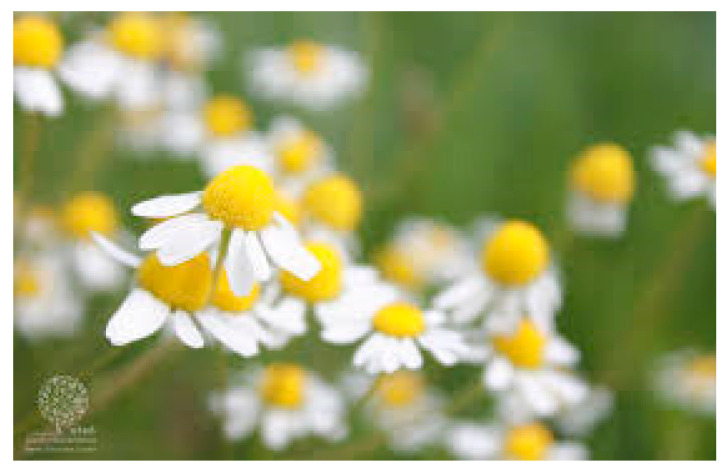
*Matricaria chamomilla*. Adapted from Jardim Botânico da UTAD [219].

**Figure 11 plants-13-03515-f011:**
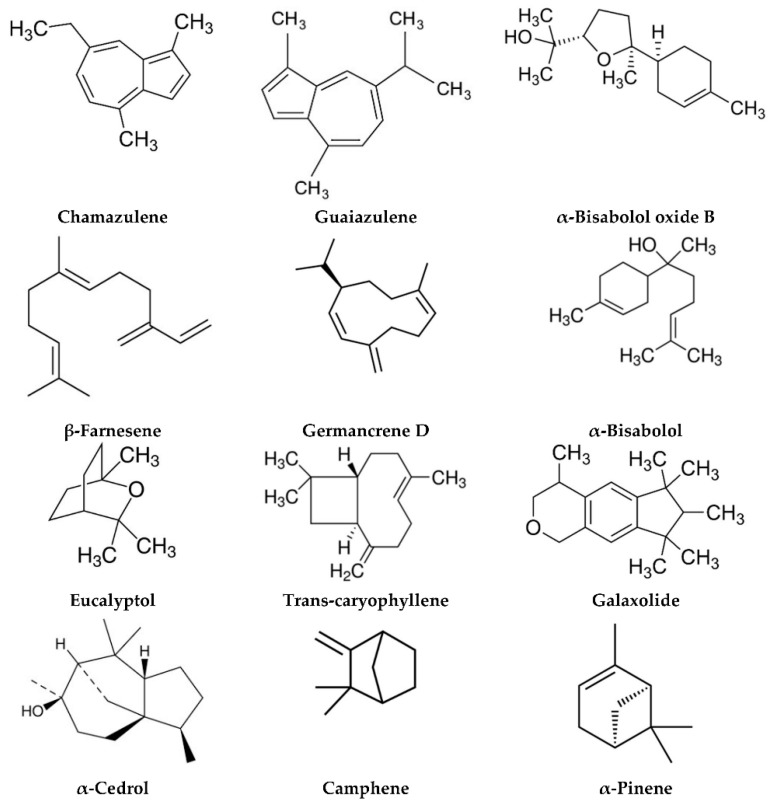
Terpenoids identified in *Matricaria chamomilla*. Adapted from Mihyaoui et al. [224].

**Figure 13 plants-13-03515-f013:**
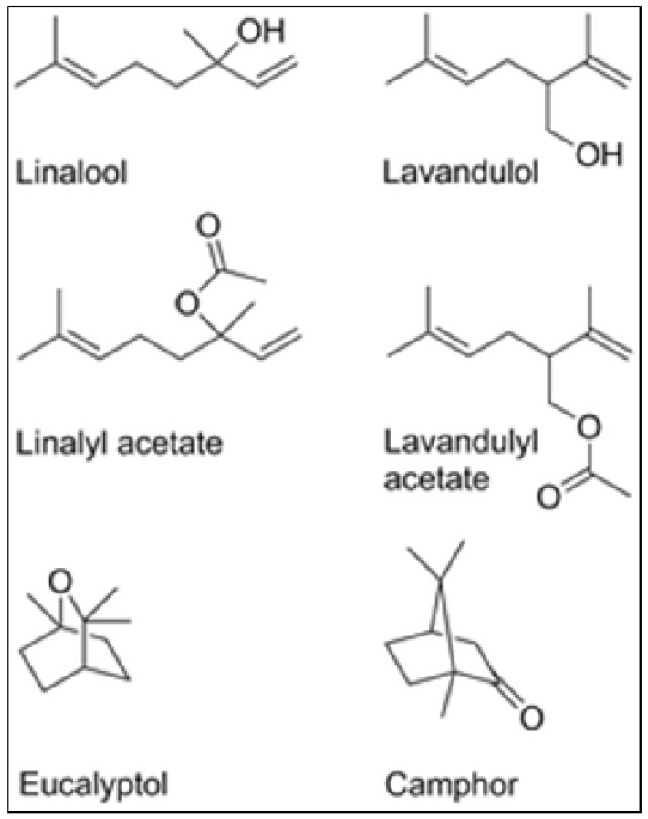
Major chemical components of *Lavandula angustifolia*. Adapted from Mahmood et al. [245].

**Figure 14 plants-13-03515-f014:**
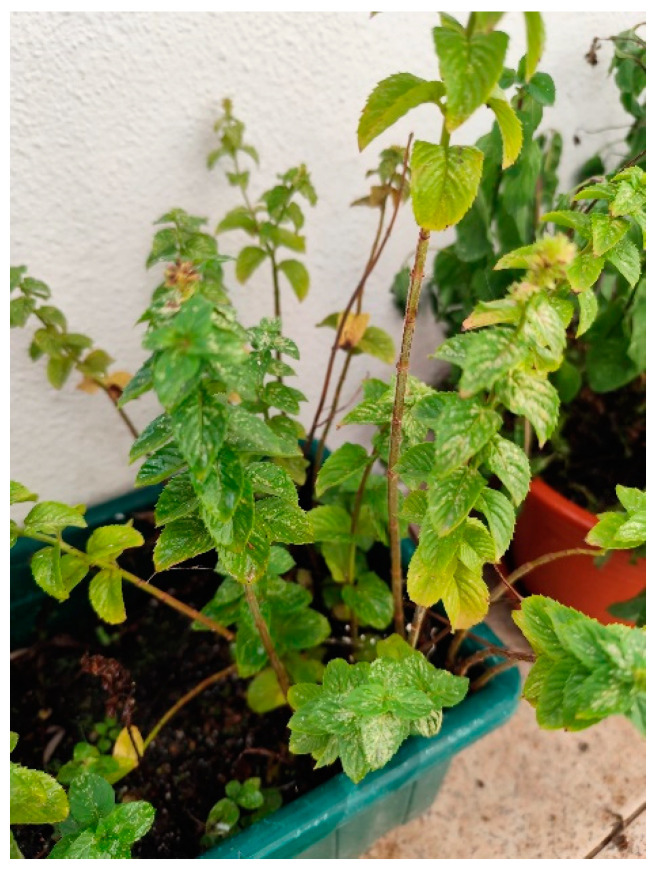
*Mentha piperita*.

**Figure 15 plants-13-03515-f015:**
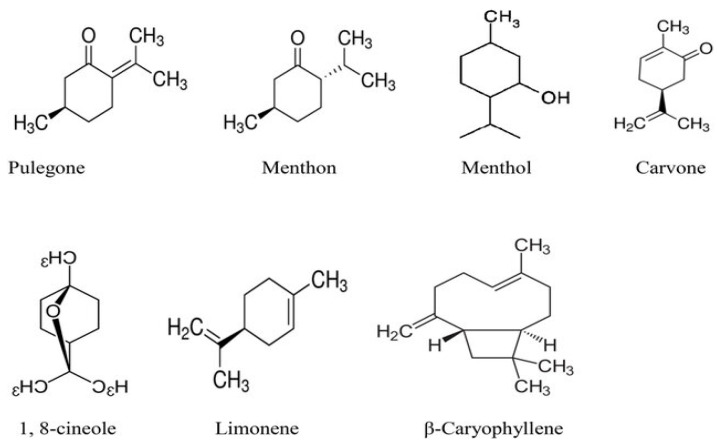
Active compounds of *Mentha piperita*. Adapted from Singh et al. [280].

**Figure 16 plants-13-03515-f016:**
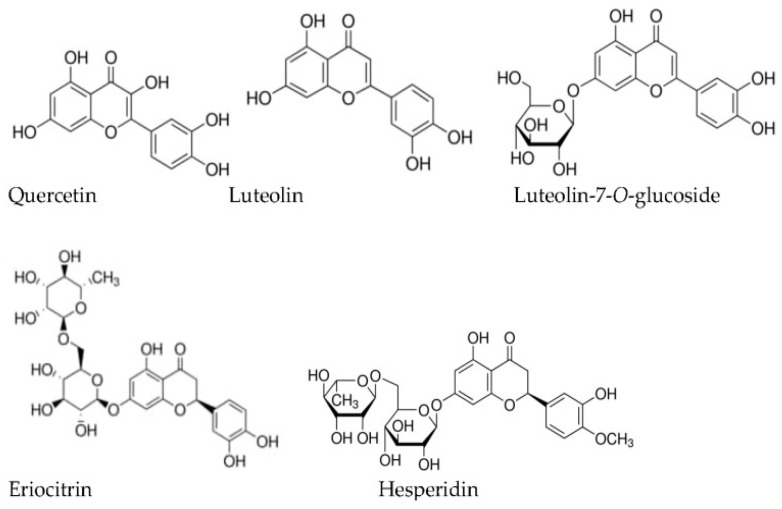
Major flavonoids of peppermint extracts (*Mentha* × *piperita*). Adapted from Hudz et al. [278].

**Figure 17 plants-13-03515-f017:**
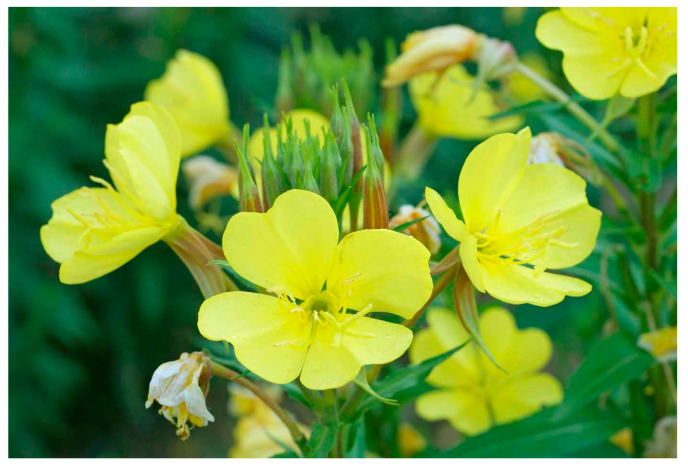
Evening Primrose. Adapted from Applewood Seed [291].

**Figure 18 plants-13-03515-f018:**
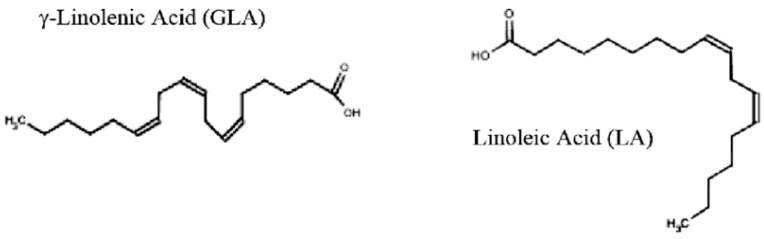
Major chemical active compounds in evening primrose oil. Adapted from Dodov et al. [301].

**Table 1 plants-13-03515-t001:** Summary of Medicinal Plants and Their Preparations for Pruritus Relief.

Plant Name	Country/Region	Common Name	Active Compounds	Mechanism of Action	Pruritus Application	Safety and Side Effects	Parts Used	Form Available	Mode of Usage/Preparation
*Aloe barbadensis*	Tropical/Subtropical	Aloe vera	Polysaccharides Flavonoids	Hydrates skin Reduces inflammation Antimicrobial	Xerosis-induced pruritus	Generally safe, mild skin irritation possible	Gel	Gel	Apply the gel directly to the affected skin
*Calendula officinalis*	Europe, North America	Marigold Calendula	Flavonoids Saponins	Anti-inflammatory Promotes wound healing	Dermatitis-associated pruritus	Safe, but avoid use on open wounds without dilution	Flowers	Oil Extract Hydrolate	Infuse flowers in oil or use in ointments Use calendula hydrolate as a spray
*Curcuma longa*	India, Southeast Asia	Turmeric	Curcuminoids Turmerone	Anti-inflammatory Antioxidant	Histamine-independent pruritus (e.g., chronic pruritus)	Safe, but can cause staining and mild skin irritation	Rhizome	Powder Cream	Apply the paste of powder and water Use in creams for topical application
*Glycyrrhiza glabra*	Europe, Asia	Licorice	Glycyrrhizin Flavonoids	Reduces inflammation Soothes skin irritation	Allergic or inflammatory pruritus (e.g., eczema)	Generally safe, high concentrations may cause irritation	Root	Powder Extract Hydrolate	Use root powder or extracts in creams or infusions for topical application Use licorice hydrolate as a toner
*Lavandula angustifolia*	Mediterranean	Lavender	Linalool Linalyl acetate	Reduces inflammation Calms skin	Stress-induced or inflammatory pruritus	Generally safe, but can cause allergic reactions in some individuals	Flowers	Essential oil Hydrolate	Use essential oil diluted in a carrier oil for topical application Use lavender hydrolate as toner
*Matricaria chamomilla*	Europe, Asia	Chamomile	Chamazulene Apigenin	Anti-inflammatory Soothes skin irritation	Cytokine-mediated pruritus (e.g., atopic dermatitis)	Safe, but can cause allergic reactions in some individuals	Flowers	Essential oil Hydrolate Tea	Apply cooled tea as a compress Use chamomile essential oil or hydrolate
*Mentha piperita*	Europe, North America	Peppermint	Menthol Flavonoids	Provides cooling sensation Reduces itching	Urticaria and itch from skin conditions	Safe, but can cause skin irritation if not properly diluted	Leaves, oil	Essential oil Hydrolate	Dilute essential oil; apply topically for a cooling effect Use peppermint hydrolate
*Oenothera* *biennis*	North America, Europe	Evening Primrose	Gamma-Linolenic Acid	Reduces inflammation Supports skin barrier function	Atopic dermatitis-related pruritus	Safe, but can cause mild gastrointestinal issues if ingested	Seeds (oil)	Oil	Apply oil directly or add to creams

**Table 3 plants-13-03515-t003:** Utilization of turmeric in pruritus.

Type of Study/Pathology	Dosage of Curcumin/ Curcuminoids	Principal Outcomes	Publication
Uremic pruritus in hemodialysis patients	66.3 mg per day (ingested)	decrease in pruritus scores in the turmeric group compared with the placebo	[184]
Sulfur mustard-induced chronic pruritus	950 mg per day (ingested)	reduction in substance P in the curcumin group compared to placebo increase in superoxide dismutase, glutathione peroxidase, and catalase in the curcumin group 30% reduction in reduction in pruritus severity in the curcumin group	[140]
Antipruritic effect of itch cream in dermatological disorders	‘Itch cream’ containing 16% turmeric (topical)	improvement in subjective pruritus severity, clinical assessment, and well-being mild skin irritation or burning at the site of application in 3 of 25 cases in the treatment group	[185]
Antipruritic effect of curcumin on histamine-induced itching in mice	Curcumin solution in Vaseline (topical)	histamine-induced itching was blocked by topical application of curcumin in a concentration-dependent manner histamine-induced discharges of peripheral nerves were reduced by the application of curcumin, indicating that curcumin acts directly on peripheral nerves curcumin and blocks the histamine-induced inward current via activation of TRPV1	[136]
Contact dermatitis as an adverse reaction to some topically used European herbal medicinal products	Cream Herbavate^®^ + *C. longa* (topical)	alleviation of itching, thickening, scaling, and erythema only 5 in 150 cases reported side effects	[186]

**Table 5 plants-13-03515-t005:** Different forms of administration and dosage of *Mentha piperita*.

Internal Use	External Use
Infusion: one dessert spoon per cup, 2 or 3 cups per day. Nervous people and/or people prone to insomnia should drink the most diluted infusion.	Liniment: with 1 to 5% essential oil.
Tincture (1:5): 50 drops, 1 to 3x/d.	
Essential oil: in a cup of infusion, 1 to 3 drops in a suitable dilution vehicle (a lump of sugar, alcoholic, or oily solution), 2 or 3x/d, with an average daily dose of 2 to 9 drops.	
Capsules: with 25 to 50 mg of essential oil, 1 to 2x/d.	
Suppositories: with 0.1 to 0.4 g of essential oil per suppository, 2 or 3x/d.	
Dry or wet inhalations: 5 to 10 drops of essential oil in 0.5 l of hot water.	
Aerosol: for 50 mL of appropriate diluent, 1 to 2 g of essential oil.	

x/d—Times per day.

**Table 6 plants-13-03515-t006:** Doses recommended by ESCOP (Portuguese acronym, English translation: European Scientific Cooperation in Phytotherapy, based in Elburg, Netherlands).

Internal Use	External Use
Infusion: 1.5 to 3 g of leaves in 150 mL of water, 3x/d.	Liquid or semi-solid preparations: - as analgesic, anesthetic, or antipruritic: 0.1 to 1% p/p essential oil; - as rubefacient: 1.5 to 16% p/p essential oil. - as cooling spray: 1–2% diluted in hydrolate.
Tincture (1:5, in 45% alcohol): 2 to 3 mL, 3x/d.	
Essential oil: - Digestive disorders: 0.2 to 0.4 mL, 3x/d, in oily, alcoholic suspension or enteral tablets; - Irritable colon: 0.2 to 0.4 mL, 3x/d, in capsules or enteral tablets; - Respiratory problems: inhalations, 3 to 4 drops in hot water or, in tablets, 2 to 10 mg.	
Hydrolate: 5–15 mL diluted in water or tea, up to 3x/d (if labeled safe for ingestion).	- Skin mist or compress: Apply directly to relieve pruritus, inflammation, or cooling. - Hair rinse: Soothe scalp irritation and itchiness.

p/p—percent weight by weight; x/d—Times per day.

## Data Availability

The data that support the findings of this study are available on request from the corresponding author, [S.G.].

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
