# Peer review of "Soothing the Itch: The Role of Medicinal Plants in Alleviating Pruritus in Palliative Care"

_plants, 2024, doi:10.3390/plants13243515_

Round 1
Reviewer 1 Report
Comments and Suggestions for Authors
The article titled "Soothing the Itch: The Role of Medicinal Plants in Alleviating Pruritus in Palliative Care" by Sara Gonçalves and collaborators is a valuable study on the potential applications of medicinal plants in the treatment of pruritus in palliative care. The authors conducted a detailed analysis of eight plants (including chamomile, aloe vera, lavender, turmeric, and licorice), focusing on their active compounds, mechanisms of action, clinical efficacy, and safety. The publication is based on an extensive body of literature, combining data from in vitro studies, animal models, and clinical trials, highlighting the interdisciplinary nature of the work.
One of the main strengths of the article is its holistic approach to the treatment of pruritus. The authors emphasize the significance of natural medicine as a complement to traditional methods, aligning with current trends in integrative medicine. They addressed not only the physical symptoms of pruritus but also its impact on patients' psychological well-being, which is particularly important in palliative care. The study also provides practical information on the use of the analyzed plants, such as preferred application forms (gels, creams, oils) and potential adverse effects.
However, a notable weakness is the lack of standardization in the methods discussed and the diversity of available data, which complicates the clear assessment of the effectiveness of specific therapies. While the authors rightly highlight the need for further clinical research, the article lacks a detailed discussion of the limitations of the current knowledge and specific directions for future studies. Additionally, some sections, such as the mechanisms of action of the plants, could benefit from more detailed explanations and support from a broader range of experimental data.
Certain issues also merit consideration by the authors. Was the possibility of interactions between the discussed plants and standard drugs used in palliative care examined? What are the potential challenges in incorporating these therapies into daily medical practice, especially regarding resource availability and costs? Furthermore, it would be worthwhile to expand on the ethical and cultural aspects of using plant-based medicine, particularly in diverse patient groups.
In conclusion, the article provides valuable insights into the use of medicinal plants for the treatment of pruritus in palliative care while emphasizing the need for further research to confirm the efficacy and safety of these therapies. It is a significant contribution to the development of complementary and integrative medicine, though it requires certain additions to fully realize the potential of the topic discussed.
Author Response
Comment 1: However, a notable weakness is the lack of standardization in the methods discussed and the diversity of available data, which complicates the clear assessment of the effectiveness of specific therapies. While the authors rightly highlight the need for further clinical research, the article lacks a detailed discussion of the limitations of the current knowledge and specific directions for future studies. Additionally, some sections, such as the mechanisms of action of the plants, could benefit from more detailed explanations and support from a broader range of experimental data.
Response 1: Thank you for your valuable feedback. We acknowledge the lack of standardization in the methods and the diversity of available data as limitations of current knowledge. The revised version will include a more detailed discussion of these limitations.
Comment 2: Was the possibility of interactions between the discussed plants and standard drugs used in palliative care examined? What are the potential challenges in incorporating these therapies into daily medical practice, especially regarding resource availability and costs? Furthermore, it would be worthwhile to expand on the ethical and cultural aspects of using plant-based medicine, particularly in diverse patient groups.
Response 2: We agree that discussing potential interactions between medicinal plants and standard palliative care drugs is an important area to explore. We will include a section addressing known interactions and the importance of consulting healthcare providers before incorporating plant-based therapies into treatment plans.
Regarding the challenges of incorporating these therapies into daily practice, we will expand the discussion to include resource availability, costs, and practical considerations in integrating plant-based medicine into routine palliative care.
We also appreciate your point on plant-based medicine's ethical and cultural aspects. We will add a section highlighting the need to respect cultural differences and ensure ethical considerations, particularly in diverse patient groups, to ensure that plant-based therapies are used appropriately and with informed consent.
Reviewer 2 Report
Comments and Suggestions for Authors
The manuscript titled Soothing the Itch: The Role of Medicinal Plants in Alleviating Pruritus in Palliative Care is well-written and provides a comprehensive review of the role of medicinal plants in managing pruritus in palliative care. The authors effectively highlight the potential of medicinal plants as complementary therapies and emphasize their holistic benefits in improving patient comfort and quality of life. The conclusions align well with the overall content, reinforcing the practical value of integrating botanical therapies into palliative care. Below are my detailed comments and suggestions for further improvement:
1. The Abstract lacks detailed information about the databases (e.g., MEDLINE, PubMed) used for the literature review. The authors should specify the selection criteria for including or excluding studies need to be clearly outlined, including parameters like study type, language restrictions, and publication date range.
2. I recommend including schematic diagram to better illustrate the mechanisms of action of the medicinal plants and their active phytochemicals.
3. I recommend including a table that organizes the following information for each plant:
Medicinal plant, Active compound(s), Mechanism of action (e.g., anti-inflammatory), Pruritus application (e.g., Cytokine-mediated pruritus), and safety profile
Author Response
Comment 1: The Abstract lacks detailed information about the databases (e.g., MEDLINE, PubMed) used for the literature review. The authors should specify the selection criteria for including or excluding studies need to be clearly outlined, including parameters like study type, language restrictions, and publication date range.
Response 1: We have updated the abstract to specify the databases used (e.g., PubMed, Web of Science, Scopus, and b-on) and outlined the study selection criteria, including study type, language restrictions, and publication date range.
Comment 2: I recommend including a schematic diagram to better illustrate the mechanisms of action of the medicinal plants and their active phytochemicals.
Response 2: Thank you for the suggestion. We have included a schematic diagram illustrating the mechanisms of action of the medicinal plants and their active phytochemicals for better visualization.
Comment 3: I recommend including a table that organizes the following information for each plant:
Medicinal plant, Active compound(s), Mechanism of action (e.g., anti-inflammatory), Pruritus application (e.g., Cytokine-mediated pruritus), and safety profile
Response 3: Thank you for the suggestion. We find that these tables are redundant since most of the information is presented in Table 1. We will, however, add the column "Pruritus application to Table 1.